# WHEN AND HOW ARE MODULAR NETWORKS BETTER?

## ABSTRACT

Many real-world learning tasks have an underlying hierarchical modular structure, composed of smaller sub-functions. Traditional neural networks (NNs), however, often ignore this structure, leading to inefficiencies in learning and generalization. Leveraging known structural information can enhance performance by aligning the network architecture with the task's inherent modularity. In this work, we investigate how modular NNs can outperform traditional dense networks by systematically varying the degree of structural knowledge incorporated. We compare architectures ranging from monolithic dense NNs, which assume no prior knowledge, to hierarchically modular NNs with shared modules, which leverage sparsity, modularity, and module reusability. Our experiments demonstrate that incorporating structural knowledge, particularly through module reuse and fixed connectivity, significantly improves learning efficiency and generalization. Hierarchically modular NNs excel in data-scarce scenarios by promoting functional specialization within the modules and reducing redundancy. These findings suggest that task-specific architectural biases can lead to more efficient, interpretable, and effective learning systems.

## 1    INTRODUCTION

Real-world learning tasks often exhibit an inherent hierarchical and modular structure, where a complex target function can be decomposed into smaller sub-functions arranged hierarchically Simon (1991). Exploiting this structure—either explicitly known or inferred during training—can significantly improve the efficiency and generalization of neural networks (NNs).

Traditional NNs typically treat target functions as undifferentiated input-output mappings, ignoring any underlying modular structure. This results in higher training costs and the need for larger datasets. Recent advances in NN architectures, particularly sparse and hierarchically modular NNs, have demonstrated the potential to overcome these challenges (Fernando et al., 2017; Rosenbaum et al., 2017; Shazeer et al., 2017; Kirsch et al., 2018; Goyal et al., 2021; Ponti et al., 2022). These architectures break the NN into sparsely connected sub-networks or modules, each learning a distinct sub-function, and arrange them hierarchically to mirror the structure of the task.

The key advantage of hierarchically modular NNs is their alignment with the natural task structure, allowing for more efficient learning. However, the challenge remains when the task structure is not explicitly known. In this study, we explore how different degrees of structural knowledge can be leveraged to enhance NN performance.

We begin by considering dense NNs with no structural assumptions, followed by random sparse NNs that assume sparsity but not its specific pattern. These fall under the category of monolithic NNs. We then examine hierarchically modular NNs, where modules are explicitly defined and organized hierarchically (Fernando et al., 2017; Ostapenko et al., 2021). In these networks, both module weights and inter-module connections are learned without assuming any prior knowledge about the specific connectivity. We further extend this exploration to modular NNs with fixed inter-module connectivity, representing cases where the exact sub-function connectivity is known.

Finally, we introduce module reusability, where the same module can be used in multiple locations within the hierarchy, reflecting the idea that sub-functions may recur throughout the task (Goyal et al., 2021; Ostapenko et al., 2022). In these cases, the NN must learn both connectivity and module selection dynamically from a shared pool of modules. We also consider a variant with fixed connectivity and module selection, representing complete structural knowledge.

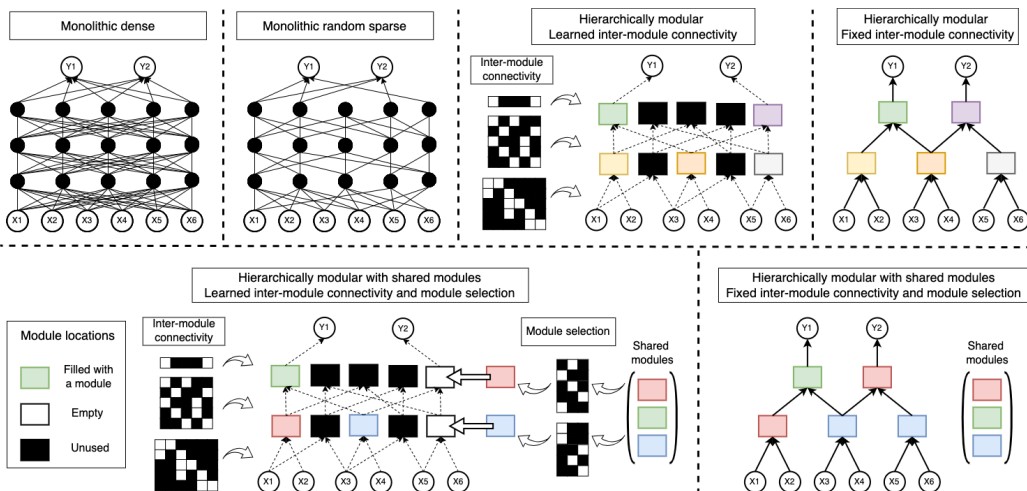

Figure 1: Overview of the models, highlighting varying levels of structural knowledge assumed at initialization. The associated task for these architectures is depicted in Figure 2 (depth 2). From top left to bottom right: Monolithic dense NN: Unknown task structure. Monolithic random sparse NN: Sparsity in task structure. Hierarchically modular NN: Modular sparsity pattern. Hierarchically modular NN (fixed inter-module connectivity): Modular sparsity with known sub-function connectivity. Hierarchically modular NN with shared modules: Modular sparsity and module reusability. Hierarchically modular NN with shared modules (fixed inter-module connectivity and module selection): Modular sparsity, module reusability, known sub-function connectivity and reuse.

All architectures studied involve learning functional components, regardless of the degree of structural knowledge incorporated. Our empirical evaluation focuses on tasks derived from Boolean functions with clear hierarchical and modular structures, allowing us to systematically analyze the effects of sparsity, modularity, and module reusability on generalization and training efficiency. Our findings demonstrate that hierarchically modular NNs with shared modules consistently outperform dense NNs, particularly in data-scarce scenarios. Further, accurately learning the inter-module connectivity and promoting functional specialization within the modules enhances their generalization performance. We also validate these findings on a visual recognition task using the MNIST dataset, showing the broader applicability of modular design principles in neural network architecture.

## 2 PRELIMINARIES

### 2.1 HIERARCHICALLY MODULAR BOOLEAN FUNCTIONS

In this work, we construct hierarchical and modular tasks using Boolean functions. A Boolean function $f : \{0,1\}^n \to \{0,1\}^m$ maps $n$ input bits to $m$ output bits. The set of gates $G$ includes $\{\wedge, \vee, \oplus\}$ (AND, OR, XOR), with edges representing direct connections.

A *function graph* for a Boolean function is represented as a directed acyclic graph (DAG) consisting of $n$ input nodes with zero in-degree, $k$ gate nodes with non-zero in-degree (associated with gates from $G$), and $m$ output nodes with zero out-degree.

A *sub-function* or sub-task corresponds to a gate node within the function graph that applies an operation on its specific inputs. Sub-functions are organized hierarchically, with outputs from certain sub-functions serving as inputs for others. These sub-functions have three fundamental properties:

1. *Input Connectivity and Separability:* Sub-functions operate on outputs from previous sub-functions or input nodes. This connectivity is sparse—each sub-function relies on a subset of preceding outputs. In our experiments, each sub-function takes exactly two inputs to maintain uniformity across graphs.

2. *Output Connectivity and Reusability:* Outputs produced by a sub-function can be reused by multiple sub-functions at higher hierarchical levels, similar to feature reuse in neural networks.

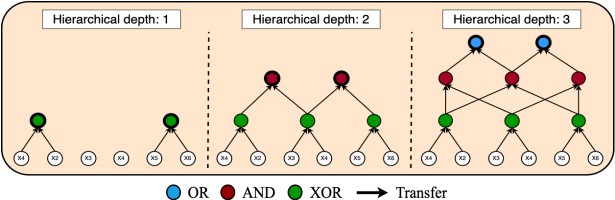

Figure 2: Function graphs with varying complexity used to generate the truth tables.

3. *Sub-Function Reusability:* The functional operation of a sub-function can be reused at multiple locations in the function graph. For instance, an XOR gate might be reused in various parts of the graph.

A task's hierarchical structure or *sub-function organization* is defined by the relationships among these underlying sub-functions.

## 2.2 NEURAL NETWORK ARCHITECTURES

In this section we describe various NN architectures used in our experiments.

**Monolithic NNs:** We consider dense multi-layer perceptrons (MLPs) and sparse MLPs. Sparse MLPs are created by random pruning to introduce sparsity, leveraging a notion of sparsity in the function graph but without its specific pattern.

**Hierarchically Modular NNs** (*modular*): The *modular* architecture is arranged in $L$ hierarchical layers, each containing $M_l$ modules, denoted as $m_l^i$, where $i$ represents the module's position within layer $l$. Each module includes a small MLP and an input selection vector $s_l^i \in \mathbb{R}^{M_{l-1}}$. The MLP learns the functional component, while the input selection vector governs inter-module connectivity by selecting inputs from the outputs of modules in the previous layer. The final outputs of the NN are selected from the set of all modules in the last layer using an output selection vector.

We explore two scenarios: 1. *Modular:* Both inter-module connectivity and module MLP weights are learned. 2. *Modular-FC:* Inter-module connectivity is fixed, inferred from the function graph, while module MLP weights are learned.

*Learning Inter-Module Connectivity:* Given outputs from layer $l-1$, $x_{l-1} \in \mathbb{R}^{M_{l-1}}$, the Sigmoid function is applied to $s_l^i$ to produce a score for each potential input, selecting the top-$k$ inputs. We use the straight-through estimator (Bengio et al., 2013) to propagate gradients through non-differentiable selections, facilitating effective learning of inter-module connectivity. We fix $k = 2$ to maintain structural consistency, which also prevents module collapse and optimizes module utilization (Goyal et al., 2021; Ostapenko et al., 2022). See Appendix A for additional implementation details and Appendix C.1 for experiments related to learning inter-module connectivity.

**Hierarchically Modular NNs with Shared Modules** (*modular-shared*): The *modular-shared* architecture extends the *modular* architecture, treating module positions as slots filled by modules from a shared pool of $M$ modules. Each slot has an input selection vector $s_l^i \in \mathbb{R}^{M_{l-1}}$ and a module selection vector $v_l^i \in \mathbb{R}^M$, which determines the module used in that slot. The final outputs are selected from the set of slots in the last layer using an output selection vector.

We explore two scenarios: 1. *Modular-shared:* The network learns both inter-module connectivity and module selection dynamically, alongside module weights. 2. *Modular-shared-FCMS:* Connectivity and module selection are fixed, with only the module weights being learned.

*Learning Inter-Module Connectivity and Module Selection:* Outputs from slots at layer $l-1$, $x_{l-1} \in \mathbb{R}^{M_{l-1}}$, are selected as inputs similarly to the previous architecture. The chosen inputs are passed to a module determined by the selection vector $v_l^i \in \mathbb{R}^M$, which is first transformed via a Softmax function to assign probabilities to each module in the shared pool. The module with the highest probability is selected (top-1). The straight-through estimator is used to compute gradients for both $s_l^i$ and $v_l^i$ (see Appendix A).

Hereafter, "hierarchically modular NNs" refers to both *modular* and *modular-shared* unless otherwise specified.

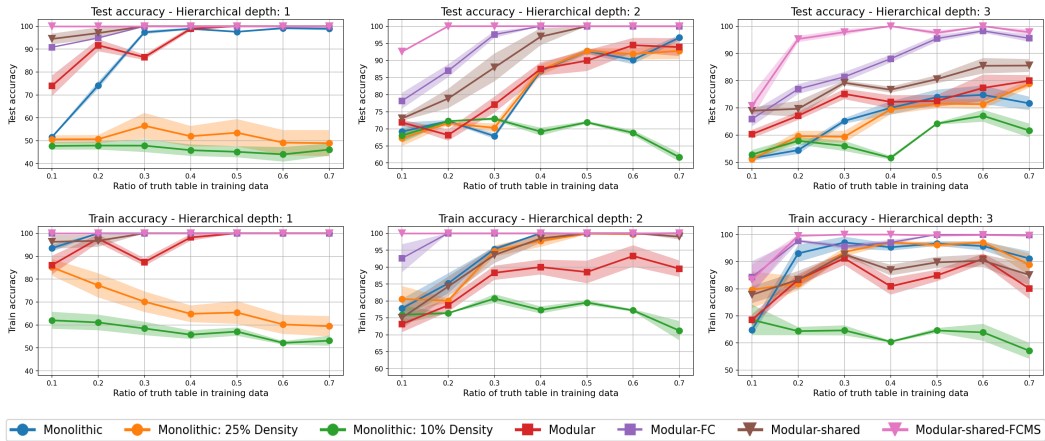

Figure 3: Test and train accuracy of different NNs relative to training size. For each datapoint, we report the mean and combined standard error (shaded region).

# 3 LEARNING MODULAR TASKS BASED ON BOOLEAN FUNCTIONS

We evaluate the performance of different NN architectures by learning Boolean functions represented as function graphs. The truth table derived from these graphs serves as the dataset. Our evaluation focuses on the models' generalization and learning efficiency when only a fraction of the truth table is available for training.

**Experiment Details:** We use three function graphs, each with 6 input nodes and 2 output nodes, as depicted in Figure 2. The complexity is controlled by the number of hierarchical levels—greater depth implies increased dependence on intermediate sub-functions, making the task more intricate.

The NN architectures are trained on different portions of the truth table, with training sizes ranging from 0.1 to 0.7 of the total rows. The remaining rows are split evenly between validation and test sets. For each training size, we create three random partitions of the truth table, and each partition is trained with three additional random seeds, resulting in a total of nine training runs per architecture per training size. We report the mean and combined standard error, calculated from the three separate means corresponding to the three dataset partitions. To enhance robustness, random noise from $\mathcal{N}(0, 0.1)$ is added to the inputs during training for data augmentation. All models are trained using the Adam optimizer for 1000 epochs.

To ensure consistency, *modular* and *modular-FC* architectures use the same number of modules as there are gate nodes at each level of the function graph (Mittal et al., 2022). Similarly, *modular-shared* and *modular-shared-FCMS* architectures align the number of slots with the number of gate nodes, and the count of shared modules matches the number of distinct gates in the graphs. Appendix E demonstrates that varying the number of modules or slots has minimal impact on performance. All module MLPs have a uniform structure, with 2 input units, a hidden layer of 12 units, and 1 output unit. Monolithic NNs are configured with 1, 3, or 5 hidden layers to match the depth of the modular NNs, with each hidden layer containing 36 units. Further architectural and training details can be found in Appendix A.

## 3.1 GENERALIZATION PERFORMANCE

**Comparing Architectures:** Figure 3 shows the generalization performance of different NN architectures relative to the training size.

1. *Monolithic Dense vs. Monolithic Sparse:* The performance of monolithic NNs is heavily influenced by parameter count. As sparsity increases, test accuracy declines significantly.

2. *Modular vs. Monolithic: Modular* NNs possess only 30%, 7.1%, and 5.9% of the weights compared to monolithic dense NNs for functions with depth 1, 2, and 3, respectively. They outperform monolithic sparse NNs with similar parameter counts (see Appendix Table 1 for details). As the

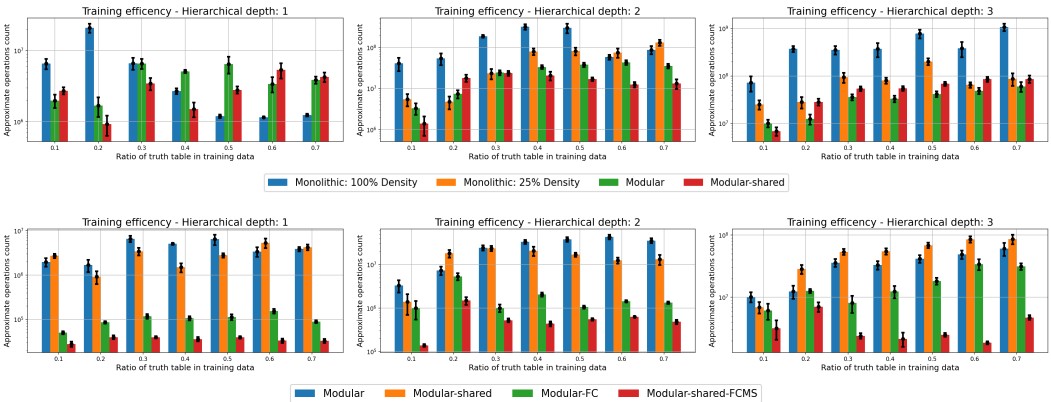

Figure 4: FLOPs required to train various NNs as compared to the ratio of truth table available.

parameter count of monolithic NNs increases, their test accuracy converges with that of *modular* NNs, indicating that prior knowledge of modular sparsity is most beneficial when parameter counts are similar.

3. *Modular-Shared vs. Monolithic and Modular:* *Modular-shared* NNs consistently outperform both monolithic and *modular* NNs, benefiting from module reuse that artificially increases the number of samples per module and leads to efficient learning of sub-functions. For depth-1 and depth-2 functions, *modular-shared* NNs significantly outperform the others. However, for depth-3 functions, the performance gap narrows, suggesting that for highly complex tasks, the benefits of modularity and reuse diminish, especially with limited training data.

4. *Fixed Connectivity and Module Selection:* *Modular-FC* and *modular-shared-FCMS* NNs demonstrate superior performance compared to all other NNs. *Modular-shared-FCMS* NNs are particularly effective with minimal training data, highlighting the value of reusability and fixed structure, which reduces the complexity associated with learning both the sub-functions and their organization.

**Train Accuracy vs. Test Accuracy:** *Modular* and *modular-shared* NNs tend to have closely aligned train and test accuracy, while monolithic NNs often exhibit overfitting, particularly with limited training data. This suggests that the inherent inductive bias of hierarchically modular NNs effectively prevents overfitting by aligning learned representations with the true task structure.

**Generalization Relative to Function Complexity:** As function complexity increases, all NNs require larger training sizes for effective generalization. A minimum threshold of training data exists for each architecture, determined by its level of prior knowledge, to generalize to unseen samples. This threshold increases with complexity but decreases with greater structural knowledge, such as module reuse and fixed connectivity.

## 3.2 TRAINING EFFICIENCY

We next evaluate training efficiency based on the number of floating-point operations (FLOPs) required for training. FLOPs are calculated considering the number of training iterations to reach peak validation accuracy, training size, and the number of weights used in forward and backward passes, as well as in optimization updates. We compare architectures in two groups, including only those that generalize effectively (see Appendix B for further details).

**Comparing Architectures:** Figure 4 presents the FLOPs required for training different NNs.

1. *Monolithic Dense vs. Monolithic Sparse:* Monolithic sparse NNs achieve better training efficiency due to reduced parameter counts. There exists a range of densities for which sparse NNs match the generalization of dense NNs while reducing training costs.

2. *Modular vs. Monolithic: Modular* NNs use only 30%, 7.1%, and 5.9% of the weights of monolithic dense NNs for depths 1, 2, and 3, respectively, leading to improved training efficiency. However, with sufficient training data, monolithic dense NNs can converge more quickly for simpler tasks, ultimately showing better efficiency, as seen for the depth-1 and depth-2 functions at high

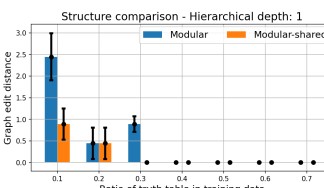 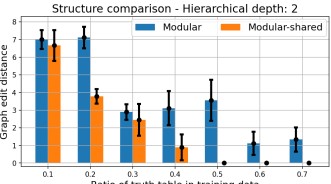 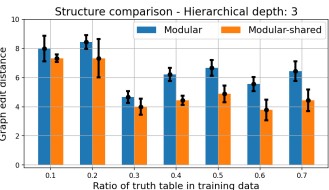

Figure 5: Minimum graph edit distance between the learned connectivity in *modular* and *modular-shared* NNs and the ground truth function graphs.

training sizes. Monolithic sparse NNs (25% density) can also match the generalization performance of *modular* NNs with similar training efficiency but lack the convergence speed-up seen in dense NNs, likely due to reduced parameterization. See Appendix B for NN convergence performance.

3. *Modular-Shared vs. Monolithic and Modular: Modular-shared* NNs consistently demonstrate superior training efficiency compared to monolithic dense NNs, although dense NNs eventually catch up with increasing training size for simpler tasks. *Modular-shared* NNs require similar FLOPs to monolithic sparse and *modular* NNs but outperform them in generalization.

Also, hierarchically modular NNs do not achieve faster convergence with larger training size – possibly because they need to explore and determine the inter-module connectivity.

4. *Fixed Connectivity and Module Selection: Modular-FC* and *modular-shared-FCMS* NNs achieve the highest training efficiency. *Modular-shared-FCMS* NNs perform particularly well, underscoring the computational advantage of focusing on learning only the sub-functions as compared to learning the sub-functions along with exploring and determining their organization. Further analysis in Appendix D shows that structural parameters require higher learning rates than module MLPs.

**Efficiency Relative to Function Complexity:** As function complexity increases, training all architectures requires more operations. For low-complexity functions, monolithic dense NNs are more efficient compared to others. However, as complexity grows, NNs with prior structural knowledge (e.g., sparsity, modularity, and reuse) achieve better efficiency. This improvement is directly tied to their ability to generalize effectively as compared to dense NNs with increasing task complexity. *Modular-FC* and *modular-shared-FCMS* NNs particularly benefit from prior knowledge of connectivity and module selection, significantly enhancing training efficiency.

### 3.3 FACTORS INFLUENCING GENERALIZATION IN MODULAR NETWORKS

In this section, we analyze two key factors influencing the generalization of *modular* and *modular-shared* NNs: learning the sub-function organization through inter-module connectivity and achieving functional specialization within modules, particularly under limited data conditions.

**Learning the Sub-Function Organization:** Unlike monolithic dense NNs, which directly learn input-output mappings, *modular* and *modular-shared* NNs need to identify the underlying task structure to perform effectively. We measure how well these NNs capture the true task structure using minimum graph edit distance (Abu-Aisheh et al., 2015), comparing the learned inter-module connectivity to the ground truth function graph.

The learned connectivity is represented as a graph with NN input units, output units, and all modules as nodes. The input and output nodes must match, while modules can align with any gate node at the same hierarchical level, ensuring permutation invariance. Figure 5 shows that *modular-shared* NNs consistently achieve lower graph edit distances compared to *modular* NNs, indicating a closer match to the ground truth. Notably, the graph edit distance is zero when both *modular* and *modular-shared* NNs achieve 100% train and test accuracy.

**Learning the Underlying Sub-Function:** The superior generalization of *modular-shared* NNs, particularly with lower truth table ratios, suggests an advantage due to module reuse across multiple locations, enabling modules to learn sub-functions more effectively with fewer samples. To quantify functional specialization, we use a metric based on Pearson's correlation coefficient between module outputs and truth table outputs for a specific sub-function.

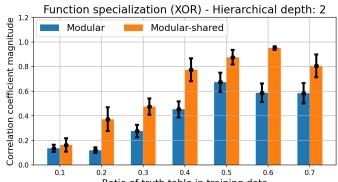
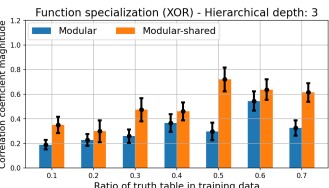

Figure 6: Magnitude of the correlation coefficient between NN module output and the XOR truth table output. Larger values indicate greater functional specialization.

Let $X$ represent all truth table rows for a specific sub-function. We collect the corresponding module outputs and calculate the correlation coefficient $\rho$ between these outputs and the ground truth. A higher magnitude of $\rho$ indicates greater alignment between the module's function and the ground truth sub-function.

Figure 6 shows the correlation coefficients for the XOR sub-function in the first hierarchical level of all three functions. Modules in *modular-shared* NNs exhibit consistently higher correlation with the ground truth compared to *modular* NNs, with $\rho$ values closely aligning with generalization performance.

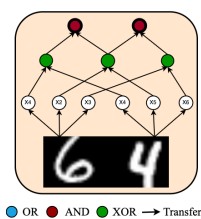
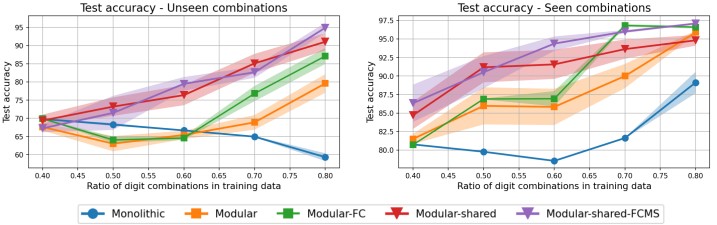

(a) MNIST task

(b) Generalization performance of various NN architectures.

Figure 7: Hierarchically modular task based on MNIST and generalization performance of various NN architectures on unseen digit combinations and seen digit combinations.

## 4 LEARNING VISUAL MODULAR TASKS BASED ON MNIST DIGITS

We present results for a modular task constructed using the MNIST handwritten digits dataset, as shown in Figure 7a. In this task, two MNIST images, each selected from digits between 0 and 7, serve as input. These images are classified into their corresponding 3-bit binary representations, which are then concatenated and passed to a Boolean task. The NNs must first classify each image independently before performing additional operations.

We vary the ratio of unique digit combinations used for training. For each training size, a random subset of all possible digit combinations is chosen, with the remaining combinations evenly divided between test and validation sets. We also present results on a test set constructed using seen digit combinations but with unseen digit images. Each training combination contains 1000 samples, while the test and validation sets contain 100 samples per combination. We also examine the effect of different numbers of samples per combination for training.

We adapted the NNs for handling image inputs: in *modular* NNs, the first hierarchical level contains MLP modules designed for image processing, with 784 input units, two hidden layers (128 and 64 units), and 3 output units. Each module processes one of the two images, and the outputs are concatenated and passed to higher levels, where modules learn Boolean functions, similar to previous experiments. In *modular-shared* NNs, two sets of shared modules are employed—one set for image processing and another for Boolean functions. The first layer contains two slots, each selecting an image-processing module from the shared pool, with outputs concatenated and passed to higher layers that use shared Boolean modules. For monolithic NNs, the input size was increased to $784 \times 2$, and the architecture was adjusted to match the depth and number of hidden units in the *modular* NN. All models were trained for 200 epochs using the Adam optimizer, with three dataset splits and three different seeds for each split (see Appendix Section A for additional details).

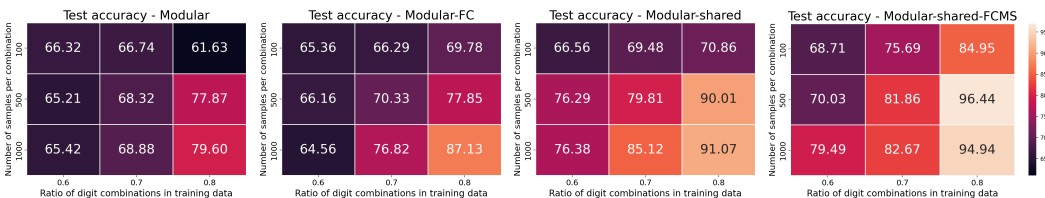

Figure 8: Generalization performance of different NNs trained on the hierarchically modular MNIST task, evaluated across varying proportions of digit combinations used for training and different numbers of samples per combination. Plots show results for: a. *modular* NN, b. *modular-FC* NN, c. *modular-shared* NN, and d. *modular-shared-FCMS* NN.

**Generalization on Unseen Digit Combinations:** Figure 7b shows the generalization performance on both unseen and seen digit combinations. At a training size of 0.4, all NNs show random test accuracy. As the training size increases, *modular*, *modular-shared*, *modular-FC*, and *modular-shared-FCMS* NNs start to generalize at different rates, while monolithic NNs do not improve and their test accuracy declines. This decrease may be due to over-fitting on seen digit combinations given that, for larger training sizes, the test accuracy of monolithic NNs on seen digit combinations improves.

The ability to generalize to unseen combinations, known as combinatorial generalization, is a persistent challenge for monolithic NNs (Keysers et al., 2019; Csordás et al., 2020). However, for Boolean tasks, monolithic NNs demonstrated the capacity to capture the underlying function and generalize as well as *modular* NNs, suggesting that simpler tasks and higher data availability can enable generalization.

*Modular-shared* NNs outperform monolithic, *modular*, and *modular-FC* NNs for the MNIST-based task. Additionally, *modular* and *modular-shared* NNs closely track their fixed-connectivity counterparts (*modular-FC* and *modular-shared-FCMS*, respectively), with small accuracy differences. This may be attributed to the large sample size used in the task, which facilitates better generalization.

Figure 8 presents the generalization performance of various NNs across different sample sizes per digit combination. We observe that *modular-shared-FCMS* NNs consistently outperform other architectures when the sample size is reduced to 500 and 100, supporting our previous hypothesis. Additionally, *modular-shared* NNs exhibit superior generalization compared to *modular* and *modular-FC* NNs, emphasizing the advantage of module reusability in low-sample training scenarios.

**Generalization on Seen Digit Combinations:** Figure 7bb illustrates the generalization performance on seen digit combinations. It is noteworthy that the validation set contains combinations distinct from both the training and test sets. The overall trends observed in previous results are consistent here. With larger training sizes, monolithic NNs demonstrate effective generalization. Notably, the *modular-FC* and *modular-shared-FCMS* architectures outperform the *modular* and *modular-shared* NNs at larger training sizes, emphasizing the advantage of leveraging predefined connectivity to facilitate learning of the complete task, encompassing both seen and unseen combinations.

**Training Efficiency:** The FLOPs required by various NNs during training are shown in Figure 9. Monolithic dense NNs show lower training costs at smaller training sizes, primarily because they reach their highest validation accuracy after just one epoch for sizes 0.4, 0.5, and 0.6, indicating limited learning. While monolithic NNs match the training efficiency of other NNs at larger training sizes, they still fall short in generalization performance.

For larger training sizes, *modular* and *modular-shared* NNs achieve training efficiencies comparable to those of *modular-FC* and *modular-shared-FCMS* NNs. This suggests that when sufficient data is available, the advantage of knowing the underlying sub-function organization has

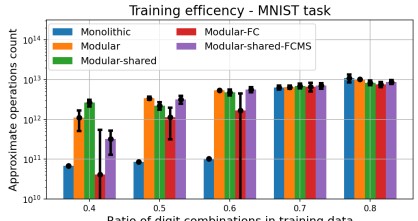

Figure 9: FLOPs required to train various NNs on the hierarchically modular MNIST task.

a limited impact on training efficiency. However, in data-scarce scenarios, this structural knowledge becomes crucial for effective training.

**Learning the Sub-Function Organization:** Similar to the Boolean function graphs, we compare the learned inter-module connectivity in *modular* and *modular-shared* NNs for the MNIST-based task by computing the minimum graph edit distance between the learned connectivity and the ground truth function graph.

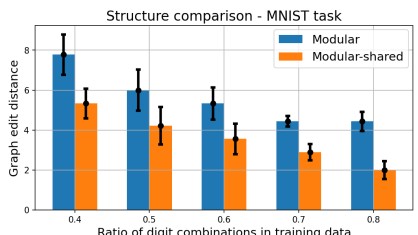

The task graph includes two input nodes (one for each image), each connected to three of six intermediate nodes in the first hierarchical level, corresponding to six output bits (three per image). These nodes are then connected to the rest of the Boolean function graph. To construct the learned connectivity graph, we represent each image module as three nodes, with incoming connections from the input nodes and outgoing connections to subsequent modules. Our results (Figure 10) indicate that *modular-shared* NNs more accurately capture the underlying connectivity compared to *modular* NNs.

Figure 10: Minimum graph edit distance between learned connectivity and ground truth task connectivity.

**Learning the Underlying Sub-Function:** Next, we evaluate the functional specialization of modules in both *modular* and *modular-shared* NNs, focusing on the image classification modules. We assess specialization using the magnitude of the Pearson correlation coefficient. Each module produces three outputs, while the ground truth classes are represented as 3-bit vectors.

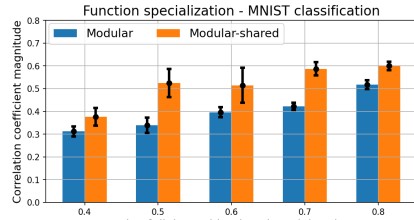

We determine the optimal permutation of the output units by maximizing the correlation coefficient and report the highest value achieved. Our results, shown in Figure 11, reveal that modules in *modular-shared* NNs exhibit a greater degree of functional specialization compared to those in *modular* NNs, reinforcing the advantage of module reusability.

Figure 11: Function specialization in image modules for *modular* and *modular-shared* NNs.

## 5 DISCUSSION

Recently, sparse NNs designed at initialization have gained attention for reducing training costs. Methods to identify these sparse NNs often involve using a portion of the training data to determine optimal sparsity patterns (Lee et al., 2019; Wang et al., 2019) or exploiting structural properties that enhance convergence and generalization (Tanaka et al., 2020; Patil & Dovrolis, 2021). However, these sparse NNs generally underperform in terms of generalization compared to dense NNs, especially at lower density levels. Accurately capturing the task structure within NNs before training remains a significant challenge.

Prior studies have examined the structure of trained dense NNs to uncover the hierarchical modularity inherent in tasks (Watanabe et al., 2018; Watanabe, 2019; Filan et al., 2021; Hod et al., 2021; Lange et al., 2022). Nonetheless, these methods do not conclusively demonstrate whether the NN's acquired structure aligns with the underlying task structure. Other works have proposed techniques to derive the task structure by training NNs and conditioning them to reveal hierarchical modularity (Malakarjun Patil et al., 2024; Liu et al., 2023).

Our work bridges these two lines of inquiry by exploring how the integration of a task's hierarchical and modular structure at initialization influences NN performance. We demonstrate that varying degrees of structural knowledge can significantly enhance generalization and training efficiency.

The concept of hierarchically modular architectures has also been applied in transfer and continual learning, where multiple tasks are learned sequentially, often assuming some level of interdependence. Previous work has utilized explicitly defined hierarchically modular NNs to address these

challenges (Terekhov et al., 2015; Veniat et al., 2020; Mendez & Eaton, 2020; Ostapenko et al., 2021). Hierarchical modularity offers advantages such as freezing modules to prevent catastrophic forgetting and reusing modules to enhance transfer performance. Similarly, multitask learning approaches have benefited from these frameworks by finding appropriate inter-module connectivity for different tasks, reusing modules for common sub-functions, and learning distinct modules for task-specific sub-functions (Devin et al., 2017; Rosenbaum et al., 2017; Maninis et al., 2019; Kanakis et al., 2020; Ponti et al., 2022).

Combinatorial generalization, also known as compositional or systematic generalization, involves the ability of NNs to generalize to unseen input combinations or tasks involving the same sub-functions arranged differently, as in visual question answering (Bahdanau et al., 2018; Lake & Baroni, 2018; Hupkes et al., 2020). Hierarchically modular NNs have demonstrated superior performance in these scenarios by enabling module reuse and reorganization based on specific parts of the input (Andreas et al., 2016; Hu et al., 2017; Wiedemer et al., 2024).

## 6 CONCLUSION

This work explored how varying degrees of structural knowledge about a task's hierarchical modularity impact NN generalization performance and training efficiency. Through experiments involving Boolean functions and a hierarchically modular MNIST task, we showed that networks with modularity and module reusability significantly outperform monolithic and sparse networks, particularly in data-limited scenarios. The improved performance stems from the ability of these NNs to exploit task structure effectively by learning both the specific sub-functions and their organization. Our findings emphasize the importance of explicitly incorporating task structure and modularity into NNs, indicating a promising direction for scalable and efficient learning systems.

A potential direction for future research is to explore the theoretical underpinnings of task learnability, focusing on its function graph and the extent of prior knowledge encoded in the NN architecture. Additionally, applying these architectures in the context of transfer learning could be valuable. Investigating how the similarity between pre-training and target tasks impacts both transfer performance and efficiency could provide key insights for optimizing transfer learning strategies.

### REPRODUCIBILITY STATEMENT

In Appendix A, we provide comprehensive details on dataset construction, NN architectures, implementation specifics, and hyper-parameter tuning, along with training procedures. Appendix B describes the algorithm used to compute FLOPs for different NNs. Additionally, Sections 3.3 and 4 detail the algorithms for computing minimum graph edit distance and functional specialization. The code is made available through an anonymous repository.

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

## A   IMPLEMENTATION AND TRAINING DETAILS

The anonymous GitHub repository can be accessed here: `https://anonymous.4open.science/r/modular-NNs-08E5`.

This section presents the implementation details for datasets, NN architectures, and training settings for all architectures.

### A.1   DATASET CONSTRUCTION AND HYPERPARAMETER TUNING

**Boolean Functions:** For Boolean functions, we generate a truth table of 64 rows (6 inputs). Rows are split into training, validation, and test sets based on the specified training ratio and dataset seed. Each training size has three different splits corresponding to three different dataset seeds.

**MNIST Task:** The MNIST task uses pairs of digits (0-7), resulting in 64 combinations. These combinations are divided into training, validation, and test sets, similarly to the Boolean functions. For each combination, we randomly select image pairs in the MNIST training set based on the specified sample size per combination. Test sets use image pairs from the MNIST test split. Dataset splitting is performed using three different seeds. When varying the number of samples per combination, the validation and test sets remain consistent.

**Training and Hyperparameter Tuning:** The NNs are trained using the Adam optimizer for 1000 epochs for Boolean functions and 200 epochs for the MNIST task. The loss function is bitwise cross-entropy with Sigmoid activation. A grid search over learning rate, batch size, and weight decay is used to select optimal hyperparameters based on validation accuracy. We use seeds $\{40, 41, 42\}$ for dataset splits and $\{0, 1, 2\}$ for NN initialization and training. We independently tune the hyperparameters for each dataset split and training size by maximizing the validation accuracy, averaged over the three training seeds.

### A.2   MLPs AND RANDOM SPARSE MLPs

**Architecture Details:** We use MLPs with ReLU activations at the hidden layers, Sigmoid at the output layers, and Xavier weight initialization (Glorot & Bengio, 2010). Sparsity in monolithic NNs is achieved by pruning edges based on a uniform random score.

Boolean functions with depths of 1, 2, and 3 use MLPs with 1, 3, and 5 hidden layers (36 units each). The MNIST-based task uses MLPs with $784 \times 2$ input units, 2 output units, and 6 hidden layers with $256, 128, 64, 36, 36, 36$ units.

**Hyperparameter Sets:** For Boolean functions, we use learning rates $\{0.1, 0.01, 0.001\}$, batch sizes $\{4, 64\}$, and weight decay $\{0.001, 0.0001\}$. For MNIST, we test learning rates $\{0.01, 0.001\}$, batch sizes $\{128, 256, 512\}$, and weight decay $\{0.001, 0.0001\}$.

### A.3   HIERARCHICALLY MODULAR NNs

**Overall Architecture:** The architecture has $L$ hierarchical layers, each with $M_l$ modules. Each module $m_l^i$ has functional parameters (MLP) and structural parameters (input selection vector $s_l^i \in \mathbb{R}^{M_{l-1}}$). The input selection vector, initialized with values from a standard normal distribution, determines module input connectivity.

**Module Input Selection:** For a module $m_l^i$, we apply the Sigmoid function to the input selection vector $s_l^i$ to get $p_l^i$, then select the top-2 values to generate one-hot encoded binary masks $b_1 \in \{0, 1\}^{M_{l-1}}$ and $b_2 \in \{0, 1\}^{M_{l-1}}$. These masks isolate inputs from $\boldsymbol{x}_{l-1}$ using dot products, resulting in inputs $x_1(l, i) = b_1 \odot \boldsymbol{x}_{l-1}$ and $x_2(l, i) = b_2 \odot \boldsymbol{x}_{l-1}$. The straight-through estimator is used to estimate gradients.

For image modules, the Softmax function is applied to $s_l^i$ and one input image is selected. A binary mask, $b \in \{0, 1\}^2$, is generated and applied to each pixel position across the two images.

**Forward Pass:** Each module's input is processed by its MLP, and outputs are concatenated before passing to the next layer. This is repeated until the final layer, where outputs are selected from the last set of modules using input selection vectors.

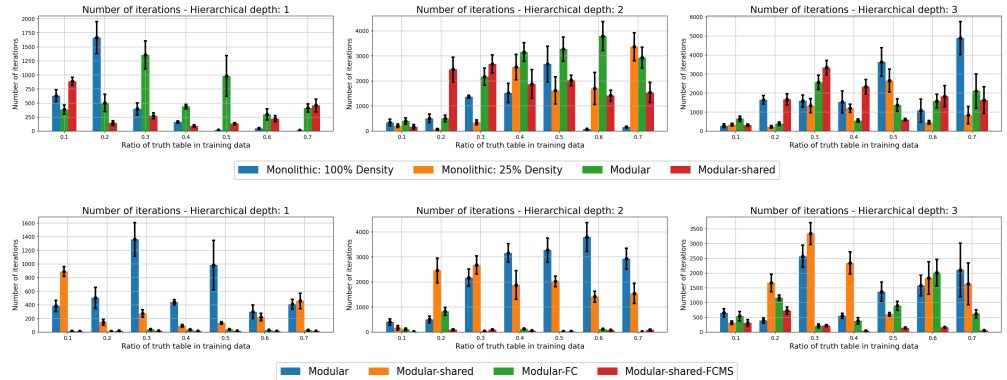

Figure 12: Number of weight updates (training iterations) for various NNs to reach peak validation accuracy on different Boolean functions as compared to the ratio of truth table available.

**Module MLP Architectures:** For Boolean sub-functions, module MLPs have 2 input units, 1 output unit, and a hidden layer with 12 units. For MNIST, module MLPs have 784 input units, 3 output units, and 2 hidden layers (128 and 64 units). Xavier initialization is used for weights.

**Hyperparameter Sets:** Hyperparameters include learning rates $\{0.1, 0.01\}$, batch sizes $\{4, 64\}$, and weight decay $\{0.001, 0.0001\}$ for Boolean functions. MNIST uses learning rates $\{0.01, 0.001\}$, batch sizes $\{128, 256, 512\}$, and weight decay $\{0.001, 0.0001\}$. Structural and functional parameters use separate learning rates, and activation function temperature $(\tau)$ for input selection vectors is also tuned $(\{1.0, 2.0, 5.0\})$.

The learning rate values tested here for Boolean functions is a subset of the one used for monolithic NNs while the batch size and weight decay values are the same. The learning rates and temperatures used here are selected from a broader range of values based on results in Appendix C.1 and D.

**Fixed Connectivity:** In fixed inter-module connectivity, input selection vectors are fixed, and gradients are not computed for them. A single learning rate is used for functional parameters, and the hyper-parameter sets tested are consistent with the setup for monolithic NNs.

A.4 HIERARCHICALLY MODULAR NNS WITH SHARED MODULES

**Overall Architecture:** The architecture consists of $L$ layers with $M_l$ slots, filled by modules from a shared pool of $M$ modules. Each slot has an input selection vector $(s_l^i)$ and a module selection vector $(v_l^i)$. Both vectors are initialized randomly with samples from the standard normal distribution.

**Input and Module Selection:** Input selection follows the same procedure as for standard hierarchically modular NNs. For module selection, the Softmax function is applied to $v_l^i$ to select a module from the pool and a binary mask, $b \in \{0, 1\}^M$ is constructed. The inputs to the slots are passed through all $M$ modules, and the slot output is computed using a dot product between the module outputs and the binary mask. The straight-through estimator is used for gradient calculation. For image slots, the module selection mask is applied independently at each module output position.

**Forward Pass:** Each slot processes inputs using a selected module, and the outputs are concatenated and passed to subsequent layers. The module MLP architecture, training, and hyperparameters are consistent with those used for hierarchically modular NNs.

**Fixed Connectivity and Module Selection:** In this variant, both input and module selection vectors are fixed, with no gradients computed. The same hyperparameters are used as for monolithic NNs.

B TRAINING EFFICIENCY DETAILS

This section describes the methodology used to calculate the number of floating-point operations (FLOPs) required during training across various NN architectures. The three main factors determining the FLOP count are: the number of samples or training size, the number of parameters, and the

| Task | Hierarchical depth 1 | | | Hierarchical depth 2 | | | Hierarchical depth 3 | | | MNIST Task | | |
|---|---|---|---|---|---|---|---|---|---|---|---|---|
| Number of weights | F | B | U | F | B | U | F | B | U | F | B | U |
| Monolithic | 288 | 288 | 288 | 2880 | 2880 | 2880 | 5472 | 5472 | 5472 | 442,744 | 442,744 | 442,744 |
| Modular | 100 | 100 | 86 | 232 | 232 | 206 | 358 | 358 | 323 | 220,840 | 220,840 | 217,682 |
| Modular-shared | 102 | 102 | 52 | 422 | 242 | 108 | 958 | 382 | 167 | 221,036 | 220,676 | 108,850 |
| Modular-FC | 72 | 72 | 72 | 180 | 180 | 180 | 288 | 288 | 288 | 217,652 | 217,652 | 217,652 |
| Modular-shared-FCMS | 72 | 72 | 36 | 180 | 180 | 72 | 288 | 288 | 108 | 217,652 | 217,652 | 108,808 |

Table 1: Number of weights involved in forward pass (F), backward pass (B) and gradient based update (U) of various NNs for different tasks.

number of training iterations (or weight updates) needed to reach the highest validation accuracy (i.e., early stopping). Figures 12 and 13 present the number of training iterations required by different NN architectures, while Table 1 summarizes the weights involved in forward, backward, and gradient update processes for various tasks.

In each training iteration, given a batch size $b$, there are $b$ forward passes, $b$ backward passes, and one weight update. For a dataset of size $D$ over one epoch, this results in $D$ forward passes, $D$ backward passes, and $\lfloor D/b \rfloor + 1$ weight updates. The total FLOP count is computed by multiplying the number of epochs by the operations performed during all forward passes, backward passes, and weight updates per epoch. FLOPs related to activation functions and biases are excluded from this calculation.

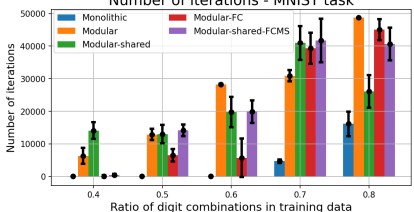

Figure 13: Number of weight updates (training iterations) for various NNs to reach peak validation accuracy on the MNIST-based task.

**Monolithic NNs:** Let $W$ be the number of weights in a dense, monolithic NN. The FLOP count for a single forward pass through the linear layers is $2 \times W$. For the backward pass, this count is $4 \times W$. Weight updates using the Adam optimizer require $18 \times W$ operations.

For random sparse monolithic NNs, the number of weights is scaled according to the network's density, and FLOP calculations follow the same approach as for dense NNs.

**Hierarchically Modular NNs:** In hierarchically modular NNs, the total FLOP count includes both operations from the forward pass through each module and those from the module input selection mechanism. For Boolean function modules, input selection involves two dot products; for image modules, it involves 784 dot products, effectively introducing additional units (2 for Boolean modules and 784 for image modules) to process the full output of the previous layer. The parameters for input selection vectors are also considered in weight updates, and the FLOP counts for the forward and backward passes incorporate $2\times$ or $784\times$ the parameters for the input selection vectors for Boolean and image processing modules, respectively. The rest of the calculations follow the previously described procedures.

In the variant with fixed inter-module connectivity, we do not include any FLOPs related to input selection and only consider the weights within the module MLPs for the FLOP calculations.

**Hierarchically Modular NNs with Shared Modules:** For hierarchically modular NNs with shared modules, each slot processes its specific inputs through all shared modules. A dot product is performed between the activated module selection vector and the outputs of all modules, increasing the number of parameters involved in the forward pass.

Let $W_m$ represent the number of weights in each module MLP, $N_s$ be the number of slots, and $M$ the number of shared modules. The number of weights used in the forward pass is $W_m \times N_s \times M$. In the backward pass, the number of active weights is reduced to $W_m \times N_s$ because the module selection mask is binary, leading to zero gradients for unselected modules in each slot.

For accounting for FLOPs associated with input selection vectors during forward and backward pass we utilize the same procedure as described for hierarchically modular NN. Module selection involves a dot product between the outputs of all modules and the module selection mask, effectively adding units (2 for Boolean modules and 3 for image modules) at the top of each slot. The forward and backward pass incorporates $1\times$ or $3\times$ the parameters for the module selection vectors for Boolean modules and image processing modules respectively.

Finally, for weight updates using the Adam optimizer, we account for the parameters in the input selection vectors, module selection vectors, and all shared modules.

The number of weights involved in the forward pass, backward pass, and optimizer updates is scaled according to the respective operations (refer to details for monolithic NNs). The total FLOP count is then obtained by multiplying these operations by the number of training epochs.

For the variant with fixed inter-module connectivity and module selection, FLOPs related to input and module selection are excluded, and only the weights in the module MLPs are considered. During the forward and backward passes, the number of active modules equals the number of slots, while weight updates are applied only to the shared modules.

## C  HYPERPARAMETER TUNING AND SELECTION

In this section, we present additional results to support the architectural and training choices for *modular* and *modular-shared* NNs. These experiments are based on the function graph shown in Figure 14, with varying proportions of the truth table used for training. Dataset details remain consistent with those described in Section A.

We perform a grid search over learning rates for both structural and functional parameters, as well as weight decay values. Tested learning rates include $0.1, 0.01, 0.001$, while weight decay values are $0.001, 0.0001$. The batch size is set to use all available training samples in a single batch, and all networks are trained for 1000 epochs using the Adam optimizer.

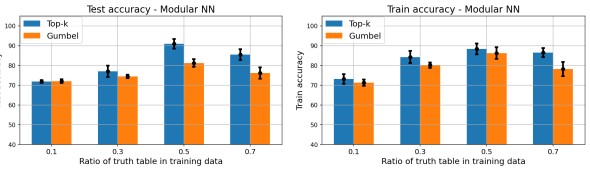

○ OR  ● AND  ● XOR  → Transfer

Figure 14: Function graph

### C.1  CONNECTIVITY AND MODULE SELECTION

In the previous section, we described the process of learning structural parameters in *modular* and *modular-shared* NNs. Here, we present experiments to justify the use of the top-$k$ operation for input connectivity and module selection in both network types.

Figure 15: Train and test accuracy of *modular* NNs as compared to the ratio of the truth table used for training. The various bars indicates the addition of Gumbel noise or direct top-$k$ for module input selection.

### C.1.1  HIERARCHICALLY MODULAR NNS

Consider the input selection vector for module $m_l^i$, denoted as $s_l^i$. The goal is to use this vector to score and select $k$ indices along with their corresponding input values for the specific module. First, the Sigmoid function is applied to the vector, yielding $p_l^i = \sigma(s_l^i)$.

Previous training methods for hierarchically modular NNs have enhanced the exploration of different connectivity patterns by adding Gumbel-distributed noise to the input selection vector before applying a normalization function. This process allows for the effective selection of the top-$k$ indices, promoting exploration during training.

We investigate a variant of this process where Gumbel noise is added to $s_l^i$ before selecting the top-$k$ indices from $p_l^i$. A grid search over the temperature parameter ($\tau$) used to normalize the vector after adding Gumbel noise was also performed to identify the best configuration. This approach aims to balance exploration and exploitation, reducing the likelihood of premature convergence to suboptimal input configurations while improving learning capability.

The results are shown in Figure 15. We observe that the standard top-$k$ selection method significantly outperforms the Gumbel noise-based variants. We also evaluate the effect of temperature on input

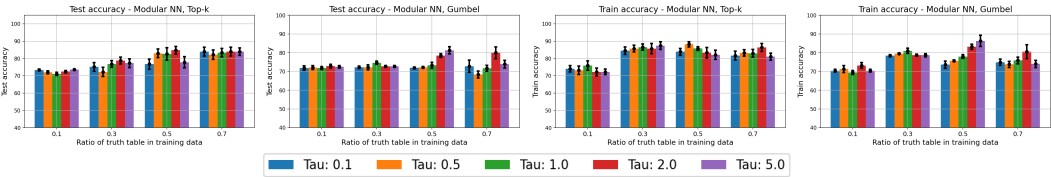

Figure 16: Train and test accuracy of *modular* NNs as compared to the temperature (tau) values used for module input selection using Gumbel noise.

selection performance. As depicted in Figure 16, higher temperature values yield better results. Increased temperature facilitates more uniform exploration of the input selection vector, contributing to improved learning outcomes.

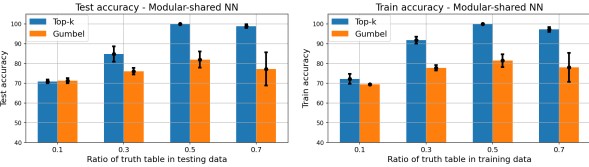

Figure 17: Train and test accuracy of *modular-shared* NNs as compared to the ratio of the truth table used for training. The various bars indicates the addition of Gumbel noise or direct top-$k$ for module input selection.

### C.1.2 HIERARCHICALLY MODULAR NNs WITH SHARED MODULES

We now present the results for the *modular-shared* architecture. Let $s_l^i$ and $v_l^i$ denote the input and module selection vectors for a given slot, respectively. The goal is to select $k$ inputs and one module for each slot. The Sigmoid function is applied to $s_l^i$ and the Softmax function is applied to $v_l^i$ to compute the selection scores.

We compare direct top-$k$ selection to a variant that uses Gumbel noise to enhance exploration. In this variant, Gumbel noise is added to the selection vectors before applying the normalization functions, which aims to avoid immediate convergence to a specific set of inputs or modules, promoting broader exploration during training.

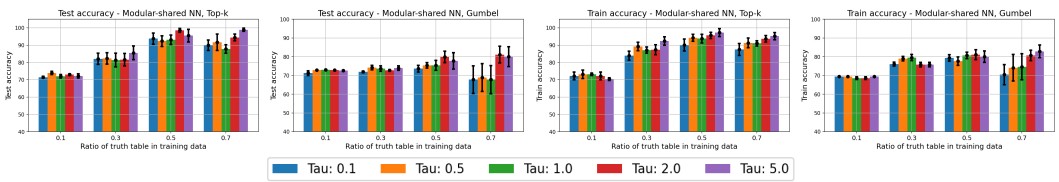

Figure 18: Train and test accuracy of *modular-shared* NNs as compared to the temperature (tau) values used for module input selection using Gumbel noise.

Figure 17 demonstrates that Gumbel noise-based variants perform worse than the standard top-$k$ selection. Furthermore, as shown in Figure 18, higher temperature values during input selection improve the model's performance, consistent with the findings for hierarchically modular NNs. This effect is due to increased exploration, preventing premature convergence and enhancing learning outcomes.

## D LEARNING RATE ANALYSIS FOR STRUCTURAL AND FUNCTIONAL PARAMETERS

We analyze the impact of learning rates on both the structural and functional parameters in hierarchically modular NNs. For both the *modular* and *modular-shared* architectures, we begin by fixing

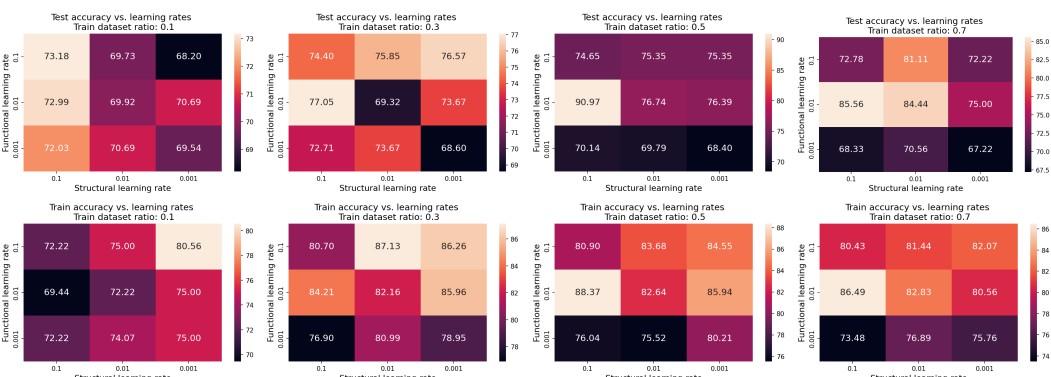

Figure 19: Test accuracy of *modular* NNs as compared to learning rates for structural and functional parameters. The various columns show the results for different truth table ratios for training.

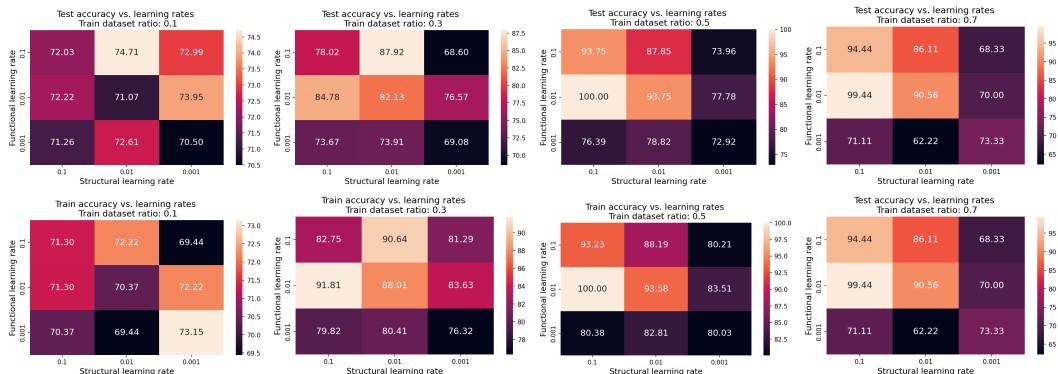

Figure 20: Test accuracy of *modular-shared* NNs as compared to learning rates for structural and functional parameters. The various columns show the results for different truth table ratios for training.

the learning rate combinations and then fine-tuning other hyperparameters, including weight decay and temperature values. The batch size is kept constant so that all available samples are used in each training iteration.

Figures 19 and 20 show the training and test accuracy for various learning rates across different training sizes for the *modular* and *modular-shared* NNs, respectively. We consistently find that the best-performing combination of learning rates is $0.1$ for structural parameters and $0.01$ for functional parameters across both architectures. This suggests that learning inter-module connectivity and module selection requires a more aggressive optimization strategy compared to learning sub-functions within the modules. Structural parameters seem to benefit from a higher learning rate, which may be due to the need for broader exploration during training.

Moreover, larger learning rates generally improve performance on tasks involving Boolean functions. Thus, for both *modular* and *modular-shared* NNs, we focus on learning rate combinations of $\{0.1, 0.01\}$. This approach reduces the complexity of the hyperparameter search space while still achieving high performance across different training data sizes.

## E  EXPERIMENTS WITH ARBITRARY MODULAR ARCHITECTURES

In this paper, we initialize *modular* NNs with the number of modules matching the function graph at each hierarchical level. For *modular-shared* NNs, we initialize the same number of slots per hierarchical level as in the function graph, while the number of shared modules corresponds to the number of distinct gates in the function graph. However, an important question arises: how effective are

these modular architectures when such structural information is unknown, and an arbitrary number of modules or slots are initialized?

In this section, we present results for scenarios where we vary the number of modules / slots at each hierarchical level, as well as the number of shared modules for the *modular-shared* architecture. These experiments provide insights into the flexibility and robustness of modular NNs under less structured initialization conditions.

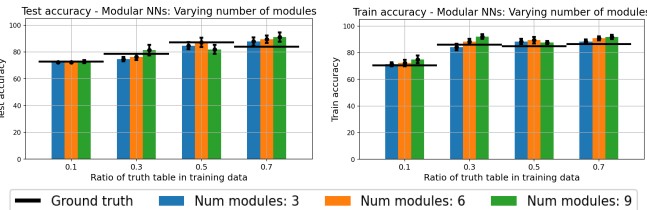

Figure 21: Train and test accuracy as compared to the number of modules defined in each layer of the *modular* NN. The black lines represent the accuracy for the model with ground truth number of modules in each layer.

### E.1 HIERARCHICALLY MODULAR NNs

In the case of hierarchically modular NNs, as illustrated in the function graph in Figure 14, the NN requires 3 modules at the first hierarchical level and 2 modules at the second. To evaluate the flexibility of the module count, we experiment with three additional architectures, where each hierarchical level is assigned $M$ modules, varying $M$ from 3 to 9.

Each architecture is trained independently using different training size ratios of the truth table, as described previously in section A. The resulting train and test accuracy values are shown in Figure 21.

Interestingly, the performance across these architectures remains comparable to the NN that uses the ground truth number of modules per hierarchical level. Furthermore, our analysis does not show a clear trend where increasing or decreasing the number of modules consistently improves performance, suggesting that the architecture is robust to variations in module count.

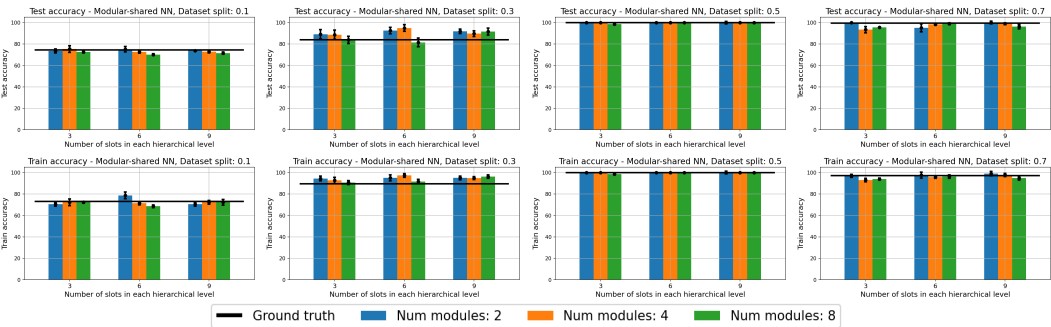

Figure 22: Train and test accuracy as compared to the number of slots defined in each layer and the number of modules in the *modular-shared* NN. The black curve represents the accuracy for the model with ground truth number of slots in each layer and the ground truth number of shared modules.

### E.2 HIERARCHICALLY MODULAR NNs WITH SHARED MODULES

For *modular-shared* NNs, we explore architectural variations along two dimensions, organized in a grid. The first dimension is the number of slots per hierarchical level, ranging $\{3, 6, 9\}$, and the second is the total number of shared modules, varying across $\{2, 4, 8\}$. This setup results in 9 distinct architectures, each defined by a unique combination of slot and shared module counts.

Each architecture is trained independently using different splits of the truth table, with varying fractions allocated for training. The results are presented in Figure 22, with plots segmented by the ratio of the truth table used for training. The horizontal black line in the figure indicates the performance of the architecture configured with the ground truth number of slots and shared modules.

As in the previous analysis, we observe that these architectures achieve similar performance to the ground truth setup.

## F ADDITIONAL RESULTS: GENERALIZATION AND TRAINING EFFICIENCY

In this section, we present additional visualizations that provide alternative perspectives on the generalization performance and training efficiency of the various architectures.

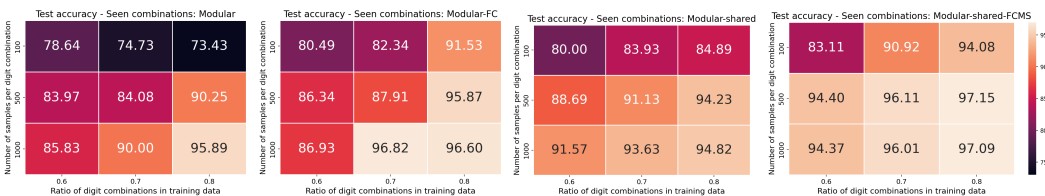

Figure 23: Generalization performance on seen combinations of different NNs trained on the hierarchically modular MNIST task, evaluated across varying proportions of digit combinations used for training and different numbers of samples per combination. Plots show results for: a. *modular* NN, b. *modular-FC* NN, c. *modular-shared* NN, and d. *modular-shared-FCMS* NN.

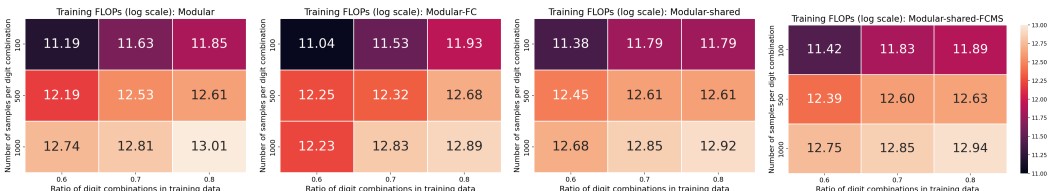

Figure 24: Training efficiency of different NNs trained on the hierarchically modular MNIST task, evaluated across varying proportions of digit combinations used for training and different numbers of samples per combination. Plots show results for: a. *modular* NN, b. *modular-FC* NN, c. *modular-shared* NN, and d. *modular-shared-FCMS* NN.

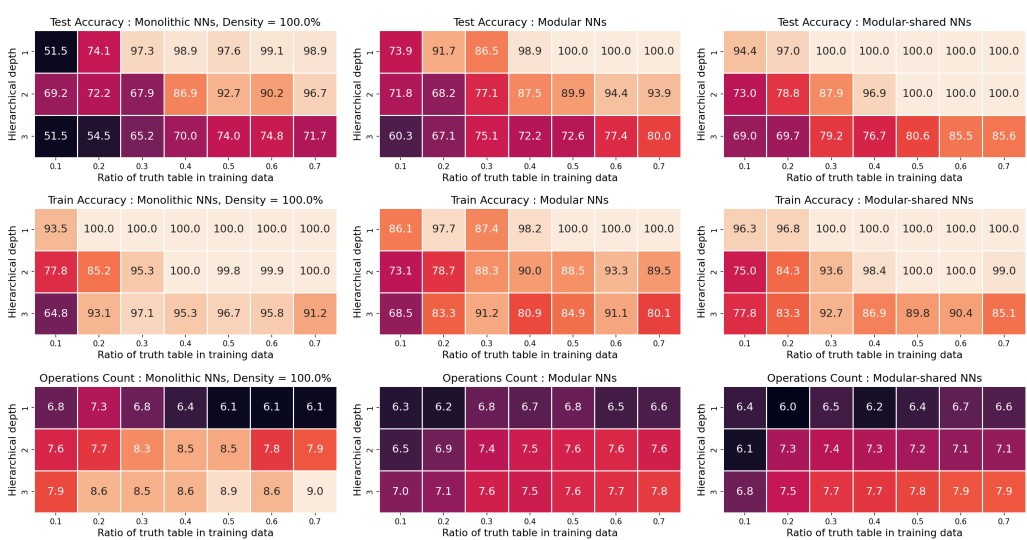

Figure 25: Test accuracy, train accuracy and FLOPs count of NNs as compared to the complexity or hierarchical depth of the Boolean function graphs and the ratio of truth table used for training. First column indicates the results for dense monolithic NNs, second column for *modular* NNs and third column for *modular-shared* NNs. We can clearly see a trend where the top right of the heatmap has better values indicating that larger training size and lower complexity functions are an easier combination to learn.