Figure 1: Function graphs used in the multi-task learning setting.

# WHEN AND HOW ARE MODULAR NETWORKS BETTER?

## 1 MULTI-TASK LEARNING

We extend our experimental setup to explore multi-task learning scenarios where each neural network (NN) architecture is trained to perform multiple Boolean functions simultaneously. This allows us to evaluate task-conditioned modularity by enabling networks to adjust connectivity and module selection based on a task identifier provided with each input sample.

### 1.1 EXPERIMENTAL SET-UP

Consider a set of $n$ Boolean function graphs (or tasks), and their corresponding truth tables $T_i$, for $i \in \{1, 2, \ldots, n\}$. Consider the experimental setup described in Section 3 of the paper. For each task $i$, we sample an $r$-fraction of rows from $T_i$, prepend a one-hot encoded task identifier $t = i$ to each row, and combine these samples from all tasks into a unified training set. This process is repeated to create the validation and test sets.

We train three architectures:

**Monolithic dense:** The input dimension of the architecture is expanded to include the task identifier. The network is then trained using standard methods.

***Modular:*** The *modular* NN is configured similarly to the single task setting, with $L$ modular layers, containing $M_l$ modules at layer $l$. Each module consists of an MLP comprising 2 input units, 12 hidden units, and 1 output unit, along with an input selection vector. The input selection vector is a two-dimensional matrix of size $n \times M_{l-1}$, where $n$ represents the number of tasks.

Given the one-hot encoded task identifier associated with each input sample, the corresponding row of the input selection vector is used to determine the module's inputs or inter-module connectivity. The input selection vector can be interpreted as the weight matrix of a single linear layer that takes the task identifier as input and generates the input selection vector specific to the module and task.

Please note that the input selection vectors for all the modules are learned or trained along with all the MLP weights.

***Modular-shared:*** Similar to the single-task modular-shared NN, this architecture replaces individual modules with $M_l$ slots in each hierarchical layer $l$ and employs a pool of $M$ shared modules. Each slot is defined by an input selection vector of size $n \times M_{l-1}$ and a module selection vector of size $n \times M$.

For each input sample, the task identifier determines the corresponding rows of these matrices, specifying both the inter-module connectivity and the module selection at each slot. The input selection and module selection vectors for all the slots are learned or trained along with all the MLP weights.

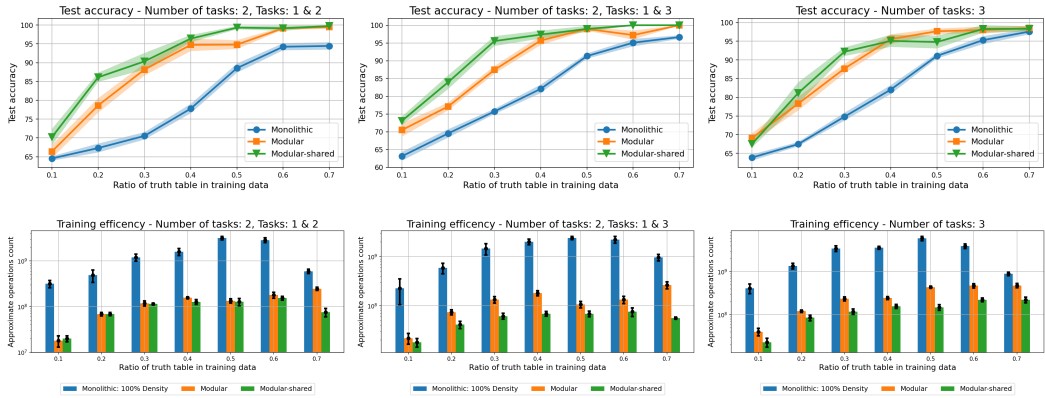

Figure 2: Test accuracy, and training efficiency results of different NNs relative to training size in the multi-task learning set-up. For each datapoint, we report the mean and combined standard error (shaded region).

## 1.2 RESULTS

We consider the three function graphs shown in Figure 1 in the multi-task setting.

**Learning two tasks (1 & 2):** First, we present results for learning two of the three tasks. Here, the monolithic dense NN consists of 8 input units, 2 output units, and 3 hidden layers with 60 units in each layer. The *modular* NN comprises 5 modules in the first layer and 4 modules in the second layer. The *modular-shared* NN consists of 4 slots in the first layer, 2 slots in the second layer, and 3 shared modules.

The test accuracy and the training efficiency are presented in Figure 2(column 1). Both *modular* and *modular-shared* NNs consistently outperform the monolithic dense NN. The accuracy gap between *modular* and *module-shared* NNs is low, likely due to the sub-function output reuse across the two tasks (e.g., 2 sub-function outputs with XOR gates are common or reused between tasks 1 and 2). *Modular* NNs with their fixed module positions, are particularly effective in capturing sub-function output reuse both across and within tasks. (See section 2 of the paper for definition)

In terms of training efficiency, both *modular* and *modular-shared* NNs require significantly fewer operations to learn the two tasks effectively. Additionally, *modular* NNs are able to match the training efficiency of *modular-shared* NNs.

**Learning two tasks (1 & 3):** Next, we present results for a multi-task setting where tasks 1 and 3 are learned simultaneously. The monolithic NN consists of 8 input units, 2 output units, and 3 hidden layers with 72 units in each layer. The *modular* NN consists of 6 modules in the first layer and 4 modules in the second layer. Finally, the *modular-shared* NN comprises 3 slots in the first layer, 2 slots in the second layer, and 3 shared modules.

The test accuracy and the training efficiency are presented in Figure 2(column 2). Both modular and modular-shared NNs again outperform the monolithic dense NN in terms of test accuracy and training efficiency. However, there is a more significant performance gap between *modular-shared* and *modular* NNs in this case. Since tasks 1 and 3 do not share sub-function outputs, the performance advantage of *modular* NNs due to common sub-function outputs is absent, leading to a larger difference in accuracy and efficiency.

**Learning three tasks:** Finally, we examine the multi-task setting where all three tasks are learned simultaneously. The monolithic NN has 9 input units, 2 output units, and 3 hidden layers with 72 units per layer. The *modular* NN is configured with 6 modules in both hierarchical layers. The *modular-shared* NN has 4 slots in the first layer, 2 slots in the second layer, and 3 shared modules.

Both *modular* and *modular-shared* NNs significantly outperform the monolithic dense NN in terms of generalization performance and training efficiency.

Across all training sizes, *modular* and *modular-shared* NNs show comparable test accuracy. This similarity can be attributed to the substantial amount of sub-function output reuse, which *modular* NNs can exploit due to their fixed module positions. However, *modular-shared* NNs consistently achieve superior training efficiency due to their reduced number of trainable parameters and the abundant sub-function operation reuse across the three tasks. (See section 2 of the paper for definition)

### 1.3 OBSERVATIONS AND IMPLICATIONS

The input or task-identifier-conditioned connectivity inherently favors hierarchically modular NNs, as their input-output function can adapt based on the task. In a single-task setting, where the input-output function is fixed, this advantage is less apparent. For this reason, we focused on a single-task or static setting earlier and found that modularity without module reusability does not improve generalization performance. However, in a multi-task setting where the task identifier is available, even *modular* NNs outperform monolithic dense NNs.