# OpenReview forum: "When and how are modular networks better?"
_ICLR.cc/2025/Conference — Submitted to ICLR 2025_

### Official Review · Reviewer_TwAX · 2024-10-30

**Soundness:** 3
**Presentation:** 2
**Contribution:** 2
**Rating:** 3
**Confidence:** 5

**Summary:**

This work compares monolithic and modular neural networks on tasks based on hierarchical composition of boolean operations. The authors highlight that in this regime with limited available data, modular systems especially with ground-truth connectivity pattern generalizes considerably better than monolithic models. The tasks considered in the work are further extended to include an image classification sub-task through the MNIST dataset. The main contributions of the work highlights that in this task regime, modular systems outperform monolithic ones in low data regime, with further benefits when the connectivity structure is known and when modules are reused. Additionally, such benefits hold even with fewer parameters, as modular networks (and reusable patterns) reduce total parameter count.

**Strengths:**

- The authors construct controlled tasks with known connectivity patterns to evaluate if modular systems work well, and if they uncover the structure of the task.
- The proposed methodology of modular systems, with structural inductive biases and reusability, outperforms monolithic systems.

**Weaknesses:**

- One of the biggest weakness of this work is that the kind of modularity and compositionality tested in this work is extremely limited. Unlike prior work [1,2] where modularity is really considered at a multi-task level, the considered work only trains each architecture on a single task which has limited and easier to discover modular structure as opposed to conditional modularity$-$ for example, if the authors trained a single model to perform a lot of different boolean tasks based on some task identification (eg. one-hot encoding or text descriptor) and saw a modular system considerably outperforming monolithic ones, that would be more of a testament of its performance benefits.

- Additionally, the authors consider an extremely toy setup of modularity where the modular structure to be leveraged is not even input dependent (eg. different inputs need to be routed differently). In contrast, they look at just static selection procedures$-$ static selection of connectivity pattern, input selection and module selection.

- The authors highlight the benefits of modular architectures on tasks with known modular connections by essentially including the known information about the task in the network architecture (eg. connectivity pattern, module details like the output should be one number, etc.). This really brings an important question of how transferable are the results from this work to more general-purpose hierarchical and modular tasks.

- Finally, the Appendix shows that monolithic models are able to catch up to modular ones with increasing training data. Combining this information with the fact that the authors only consider this very specific class of toy tasks, it is extremely unclear how transferable the conclusions are. It would be much more convincing if the authors test their approach on some more complex tasks (eg. object centric learning, multi-task or few-shot learning problems, etc.).

- Furthermore, they could try boolean functions of increasing complexity$-$ like 100 input bits with larger depth and conditioning on some one-hot vector to describe different functional forms of the ground-truth.

- The authors should also test their method on tasks that operate beyond interactions of just two bits. What about ternary interactions and other higher order mechanisms?

[1] Mittal, Sarthak, Yoshua Bengio, and Guillaume Lajoie. "Is a modular architecture enough?." Advances in Neural Information Processing Systems 35 (2022): 28747-28760.

[2] Goyal, Anirudh, et al. "Recurrent independent mechanisms." arXiv preprint arXiv:1909.10893 (2019).

**Questions:**

- Do the authors see the same kind of benefits when the modules output vectors in some latent space, as opposed to scalars?
- Have the authors considered any task that is either not synthetic or not extremely controlled (where it is easy to introduce a modular architecture that ties closely to the ground-truth), to really evaluate the benefits of modularity in a more general manner?

---

> ### Author Response · Authors · 2024-11-23
>
> We sincerely thank the reviewer for their detailed feedback and constructive suggestions. Below, we address the weaknesses (numbered W1, W2, ..) and questions (numbered Q1, Q2) raised.
>
> **W1 & W2**: To address these concerns, we extended our experimental setup to explore multi-task learning scenarios where each neural network architecture is trained to perform multiple Boolean functions simultaneously. This allows us to evaluate task-conditioned modularity by enabling networks to adjust connectivity and module selection based on a task identifier provided with each input sample.
>
> We present experiment details and results with multi-task learning in the newly added supplementary material pdf, separate from the existing appendix sections. We urge the reviewer to go through the supplementary material pdf.
>
> *Experiment finding*: Both modular and modular-shared NNs out-perform monolithic dense NN by a large margin in terms of both generalization performance and training efficiency.  Due to the limited time available during the discussion phase we were unable to generate results for all the architectures. We will have the complete set of results in the next revision.
>
> The input or task-identifier-conditioned connectivity inherently favors hierarchically modular NNs, as their input-output function can adapt based on the task. In a single-task setting, where the input-output function is fixed, this advantage is less apparent. For this reason, we focused on a single-task or static setting earlier and found that modularity without module reusability does not improve generalization performance. However, in a multi-task setting where the task identifier is available, even modular NNs outperform monolithic dense NNs.
>
> **W3**: We agree with the reviewer that the module input dimension and output dimension information is assumed to be known structural information in hierarchically modular NNs considered.
>
> The input bottleneck at the module has been previously shown to be necessary for specialization and to avoid module collapse [1,2]. If all the modules are allowed to take as input the full output of the previous hierarchical layer, then many modules would not be utilized, due to the high expressive power of individual modules. Further, such a network would be monolithic if the modules are single layered. We can add such a network (with multi-layered modules that take the entire output of the previous layer as input) as a mid-point of assumed structural knowledge between monolithic sparse and modular with 2 input modules. A key future direction can be to learn this input bottleneck without assuming the number of inputs or the module’s input dimension.
>
> The output dimension also plays a similar role in the architectures considered (1 in the case of Boolean modules and 3 in the case of image processing modules). Those are necessary for module specialization and avoiding collapse.
>
> We would like to point out that the key idea behind the paper is not to evaluate the benefits of modularity, but to understand what degree of knowledge about the task’s modularity, when incorporated into an architecture, can lead to those benefits. As pointed out by the reviewer this knowledge includes modularity, module input and output dimension, and module reusability. We show that such partial knowledge is beneficial, when compared against monolithic NNs. While extending these findings to more complex tasks and larger models is an important future direction, such exploration would need to be conducted in the context of specific tasks, the knowledge associated with those tasks, and the corresponding architectures.
>
> **W4**: We would like to point the reviewer to section 4, where we present results on the MNIST based hierarchical and modular task, and the response to W1 and W2, results on multi-task learning.
>
> For larger training sizes, the monolithic dense NNs are able to catch up with modular and modular-shared NNs in the case of Boolean functions. However, we would like to point out that this is not the case for the MNIST based task presented in section 4 (Figure 7b-1). In the MNIST based tasks we find that the monolithic dense NNs have a tendency to “over-fit” to the digit combinations present in the training data, evident by the increasing test accuracy on the seen digit combinations in Figure 7b-2.  The complexity of the task (and sub-functions) along with the number of samples play a crucial role in the performance on the unseen input combinations in both Boolean functions and the MNIST based tasks.

---

> ### Author Response · Authors · 2024-11-23
>
> **W5**: We do not fully understand the comment, however, we try our best to answer it. Please let us know if our response is not aligned with your intended comment. We believe that this comment points towards input conditioned connectivity and module selection. We have provided the results for the multi-task scenario in our response to W1 and W2.
>
> Additionally, due to the time constraints associated with the discussion period we could not present results with very complex functions. This would remain a direction for future work.
>
> **W6**: In addition to the response for W3, we would like to point out that the assumption regarding 2 input bits for each module or gate is consistent with the literature of Boolean circuits, where any Boolean function can be represented as a circuit or graph which consists of gates with two input bits. Therefore, we believe this is a reasonable assumption to make even when scaling to a larger number of input bits in the overall task, or to deeper hierarchies.
>
> **Q1**: We have not tried modular NNs with module outputs in some latent space. However, we believe that the fixed output dimension of 1 in Boolean functions and 3 in MNIST classification modules plays an important role for function specialization in the modules. Future work can consider such networks as well. Please also see the response to W3.
>
> **Q2**: We have not experimented with tasks other than the ones presented in the paper or in the newly added supplementary material pdf. Please also see the response to W3.
>
> [1] - Goyal, Anirudh, et al. “Neural Production Systems: Learning Rule-Governed Visual Dynamics.” CoRR, abs/2103.01937 (2022)
>
> [2] - Ostapenko, Oleksiy, et al. “Attention for compositional modularity.” NeurIPS'22 Workshop on All Things Attention: Bridging Different Perspectives on Attention. 2022
>
> [3] - Malakarjun Patil, Shreyas, et al. "Neural Sculpting: Uncovering hierarchically modular task structure in neural networks through pruning and network analysis." NeurIPS 2023.

---

> > ### Comment · Reviewer_TwAX · 2024-11-26
> > **Reviewer Response**
> >
> > I appreciate the authors for taking the time to provide a detailed feedback and for conducting additional experiments. However, my concern about the usability of such models in complex machine learning scenarios remains unanswered and therefore I will keep my score.
> >
> > **Multi-task setting**: While the authors show that modular systems outperform monolithic ones in the multi-task setting, it is not clear if that is the artifact of an additional supervision being provided to modular systems; which is essentially the distinction of what information decides the task and what information is input as bits. In particular, it is akin to the Modular-op system in [1] which has already been shown to perform better than non-privileged modular systems and monolithic systems. Not only are the experiments for multi-task learning very specific and synthetic, they also do not provide a fair comparison.
> >
> > **Novelty**: It is unclear what the main novelty of the work really is. [1] already shows how modular systems outperform monolithic ones when they are provided with privileged information regarding the tasks; this can be seen akin to providing structural information in a privileged manner. It seems like the authors perform similar analysis with boolean functions instead. Even beyond, the depth of analysis regarding specialization and collapse of modules is completely missing.
> >
> > **Scale of Experiments**: We have already seen across multiple architectures that developing very specific task-tailored modular systems do outperform monolithic ones, but in general-purpose tasks this advantage disappears, and even becomes more of a disadvantage. The authors show that in the specific domain of boolean functions, modular architectures outperform monolithic ones. However, it is not clear what the implications of this result are, since [1] already shows something similar across a wider array of tasks and architectures.
> >
> > I would urge the authors to consider either showing modular systems outperforming monolithic ones across a diverse array of tasks, or that modular systems can outperform monolithic ones (with or without structural information) on a specific but large-scale / real-world task.
> >
> > [1] Mittal, Sarthak, Yoshua Bengio, and Guillaume Lajoie. "Is a modular architecture enough?." Advances in Neural Information Processing Systems 35 (2022): 28747-28760.

---

> > > ### Author Response · Authors · 2024-11-27
> > >
> > > We appreciate the detailed feedback and aim to provide clarity on the objectives and experimental design of our work. Our response will highlight the unique aspects of our study, differentiate it from [1], and address concerns about multi-task learning and scalability.
> > >
> > > **1. Novelty and comparison with [1]**
> > >
> > > Our research builds upon existing work by exploring distinct approaches to modular neural network design. Key differences include variations in research questions, tasks considered and architectural approaches.
> > >
> > > ***Research Questions***
> > >
> > > Reference [1] focuses on developing benchmarks for mixture-of-experts-based models, examining how privileged input information influences routing and module specialization. Their experiments maintained consistent architectures for each task, varying only the routing input.
> > >
> > > Our study takes a different approach by systematically exploring architectural capabilities through function graph properties. We investigate modules with specific configurations, module sharing and reusability, and inter-module connectivity. While [1] emphasizes routing input selection, we concentrate on how architectural design informed by task structures can optimize neural network performance.
> > >
> > > ***Task and Network Design Differences***
> > >
> > > In [1], the task is a simple linear composition of sub-functions. Each input includes a sub-function identifier that determines the specific transformation (sub-function) applied to produce the output. The architecture features a single mixture-of-experts (MoE) layer, with modules designed to learn specific sub-functions. The routing mechanism linearly combines module outputs, with architectural variations limited to inputs to routing, such as the full input sample with the identifier, the identifier alone, or fixed connectivity.
> > >
> > > Our approach investigates more complex function graphs by utilizing multiple hierarchical levels, enabling sub-function reusability across layers, and incorporating deeper architectures for increased task complexity. We focus on architectural capabilities, exploring modular configurations, module sharing, and predefined connectivity.
> > >
> > > Mixture-of-Experts and Modular-shared Networks: The MoE layer in [1] requires independent re-learning of each sub-function. Our modular-shared network introduces an approach with "slots" functioning as full MoE layers with hard selection. We reuse the same MoE layer and modules across all slots, computing slot outputs by combining module outputs using slot-specific selection vectors and dot products. We extend the single-layer MoE architecture by introducing multiple parallel slots and hierarchical layers, enhancing functionality and flexibility.
> > >
> > > ***Analysis of Module Specialization and Collapse***
> > >
> > > Comment: “*The depth of analysis regarding specialization and collapse of modules is completely missing.*”
> > >
> > > We recognize module collapse as a well-documented challenge in modular networks. To address this, we incorporated input bottleneck strategies, as suggested by prior research ([2], [3]), enabling us to focus on the architectural capabilities and function graph properties central to this study.
> > >
> > > Additionally, we introduce metrics that independently evaluate data flow and the specific functions learned by individual modules. By disentangling inter-module connectivity from functional specialization, these metrics provide a clearer and more comprehensive analysis of network functionality.
> > >
> > > In contrast to [1], where functional specialization naturally follows task routing due to the single-layer design, our approach explicitly separates connectivity (task-based routing) from specialization (module learning). This distinction allows us to examine these aspects in greater depth across hierarchically modular networks.

---

> ### Author Response · Authors · 2024-11-27
>
> **2. Multi-task Learning**
>
> While multi-task learning is not the primary focus of this work, our experiments illustrate that the benefits of modular networks with shared modules extend to multi-task settings without performance degradation.
>
> Building on [1], which established that modular networks perform optimally when task identifiers are provided and routing is guided solely by those identifiers, we focus on architectures designed with this information. This approach allows us to examine architectural capabilities, such as modularity and module reusability, rather than revisiting findings already established in prior work.
>
> **3. Scale of Experiments**
>
> We recognize the importance of scaling experiments to larger, real-world tasks and view it as a valuable direction for future research. Our current work lays the groundwork by providing insights into modularity, module reusability, and inter-module connectivity in controlled settings.
>
> The MNIST-based modular task in Section 4 addresses a critical class of problems that require joint feature extraction and high-level decision-making. The results demonstrate a clear and substantial performance gap between monolithic dense networks and modular-shared networks, indicating some practical advantages of modular architectures.
>
> **4. Conclusion**
>
> This study builds on and extends prior work in several significant ways, contributing to the field by:
>
> - Introducing and comparing hierarchical modular networks with diverse architectural capabilities, such as shared and reusable modules.
>
> - Developing metrics to disentangle functional specialization and inter-module connectivity for a clearer understanding of network behavior.
>
> - Expanding the experimental framework to investigate complex function graphs beyond simple linear compositions.
>
> We thank the reviewer for their valuable feedback and hope this response highlights the novel contributions and broader implications of our work.
>
>
>
> [1] Mittal, Sarthak, Yoshua Bengio, and Guillaume Lajoie. "Is a modular architecture enough?." Advances in Neural Information Processing Systems 35 (2022): 28747-28760.
>
> [2] - Goyal, Anirudh, et al. “Neural Production Systems: Learning Rule-Governed Visual Dynamics.” CoRR, abs/2103.01937 (2022)
>
> [3] - Ostapenko, Oleksiy, et al. “Attention for compositional modularity.” NeurIPS'22 Workshop on All Things Attention: Bridging Different Perspectives on Attention. 2022

---

> ### Comment · Reviewer_TwAX · 2024-11-28
> **Reviewer Response**
>
> Thanks to the authors for providing a detailed response, but unfortunately my concerns weren't addressed and I outline the reasons below.
>
> **Research Questions**
>
> The research questions considered in this work have been of interest in prior literature; in particular from the lens of reusability of modules [2], communication between modules [1], and the notion of modules as slots [3] (to name a few; there are many more references for each). Given that a diversity of literature exists in this space, while I agree a systematic comparison of all these questions is a considerable novelty, I don't think this work in its current form achieves it due to the extremely specific tailored evaluation.
>
> **Task and Network Design Differences**
>
> The notion of reusability exists even in Mittal et. al and other relevant literature when they look at recurrent models. Recurrence and transformer-like architectures essentially prescribe a notion of reusability across tokens. I agree with the authors that the kind of modularity that they consider can be seen as something more general (eg. they do sub-function computations without prescribing a specific ordering which is present in RNNs), but as I mentioned before, the implications of this generality are not well described due to the lack of more complex tasks, especially since something very close to the approach already exists with temporal ordering [4].
>
> **Analysis of Module Specialization and Collapse**
>
> I find the claims made by the authors here incorrect. Yes, the information bottleneck *that heavily leverages task information* can prevent collapse but again, it is not clear what this *task information* should be in general-purpose problems. Of course if you limit each module to look at two variables, it cannot compute a function of three variables but this choice of how many variables should a module look at is highly arbitrary when it comes to a vast majority of problems.
>
> **Scale of Experiments**
>
> Apologies that I was not clear here. By scale of experiments I don't just mean more real-world and complex datasets. Such a study could be on synthetic benchmarks *where one does not have access to such privileged information of how the true solution should be structured*. For example, in the current setup there is a lot of information that is baked into modular systems and monolithic ones don't perform as well because they just don't have such an information provided to them (e.g. how a module should act on two variables only, their output should be a single bit, etc.).
>
> I would like to point out that the general problem that the authors consider is quite relevant and important for the community. In particular, concrete and systematic analysis about module reusability, inter-module communication and modular decomposition in general is important to get answers on; my main problem with the current draft is that it does not provide this systematic analysis in a more general case and only limits to a very specific application where a lot of task information is already known and then baked into these "modular" systems. I would really encourage the reviewers to conduct a more extensive empirical evaluation across modalities or different synthetic setups with varying degrees of information present about the task.
>
> [1] Goyal, Anirudh, et al. "Recurrent independent mechanisms." arXiv preprint arXiv:1909.10893 (2019).
>
> [2] Dehghani, Mostafa, et al. "Universal transformers." arXiv preprint arXiv:1807.03819 (2018).
>
> [3] Locatello, Francesco, et al. "Object-centric learning with slot attention." Advances in neural information processing systems 33 (2020): 11525-11538.
>
> [4] Goyal, Anirudh, et al. "Object files and schemata: Factorizing declarative and procedural knowledge in dynamical systems." arXiv preprint arXiv:2006.16225 (2020).

---

> ### Author Response · Authors · 2024-11-28
>
> We sincerely thank the reviewer for their constructive feedback, which will greatly assist us in improving the paper. We also deeply appreciate their acknowledgment of the novelty and contributions of our work.
>
> **Comments**
>
> 1. “*Of course if you limit each module to look at two variables, it cannot compute a function of three variables but this choice of how many variables should a module look at is highly arbitrary when it comes to a vast majority of problems.*”
>
> 2. “*Such a study could be on synthetic benchmarks where one does not have access to such privileged information of how the true solution should be structured. For example, in the current setup there is a lot of information that is baked into modular systems and monolithic ones don't perform as well because they just don't have such an information provided to them (e.g. how a module should act on two variables only, their output should be a single bit, etc.)*”
>
> 3. “*The current draft does not provide the systematic analysis in a more general case and only limits to a very specific application where a lot of task information is already known and then baked into these ‘modular’ systems.*”
>
> **Response**
>
> From these comments, we understand the reviewer's primary concern is the lack of systematic analysis in transitioning from monolithic to modular networks. Specifically, our current design or transition to modular networks made two key assumptions:
>
> -The number of inputs and outputs per module.
>
> -The depth of the hierarchically modular networks or the number of hierarchical layers.
>
> To address these concerns comprehensively, we commit to the following improvements in the final submission, should the paper be accepted:
>
> 1. Incorporating intermediate architectures that systematically vary the number of module inputs and outputs.
>
> 2. Exploring modular networks of varying number of hierarchical layers or depths.
>
> These additions will provide a more systematic evaluation of the transition to modular architectures, explicitly clarifying the impact of the assumptions in our current setup on network performance.
>
> We greatly appreciate the reviewer’s insightful comments and look forward to refining the paper accordingly.

---

> > ### Comment · Reviewer_TwAX · 2024-12-02
> > **Reviewer Response**
> >
> > I agree with the authors' outlook of my response. I would also add to it that the authors should consider additional tasks beyond the boolean setup: e.g. multi-task image classification or similar tasks where the notion of shared vs separate information is not clear; or synthetic task with a mixture of experts kind of setup without incorporating those exact biases in the architecture (e.g the input/output shape or hidden dimensions or linear / nonlinear setups). Additionally, such an analysis should be combined with at least a small analysis of how things change with increasing number of parameters just because the current models are quite small.

---

### Official Review · Reviewer_RcFR · 2024-10-31

**Soundness:** 4
**Presentation:** 4
**Contribution:** 3
**Rating:** 6
**Confidence:** 3

**Summary:**

This paper explores how incorporating structural knowledge about a learning task into a neural network can significantly improve the learning efficiency and generalisation of the model compared to a dense end-to-end model. The task explicitly focuses on Hierarchical Binary Functions. The paper demonstrates that on increasingly more Hierarchical structured tasks, using modular structured networks can improve generalisation ability while reducing training and being performant in data-limited settings, outperforming conventional dense end-to-end models.

**Strengths:**

Very straightforward and coherent paper; clearly, time was taken to set up, analyse, and present the results. Very enjoyable to read.

Although the task is simple, it highlights and showcases how incorporating the structural knowledge of the task can lead to significant performance increases.

**More specific**

Figure 1 is beneficial in understanding the different types of models explored.

Clear and well-explained experiment details. That makes sense for the hypothesis being tested.

The mean and standard deviation are reported.

Future work looks very interesting and has a neat direction to take the work.

**Weaknesses:**

The MNIST Binary task- is a little confusing at first. Please add a walkthrough of the task to the appendix to aid readers' understanding.

The abstract (lines 23-24) states how this can lead to more interpretable learning. However, the body of the paper needs to explore or mention this again. Could the authors please add a section/paragraph clarifying how this leads to more interpretable networks?

**MISC**

Line 29-30: the wrong type of citation \citep{} should be used instead, unless quoting directly, then put quote marks around the text "".


Figure 1: The Purple box is not defined in the figure. What is it?

Figure 4: The first figure does not show the Monolithic with 25% Density; this figure is also hard to read; about the text, it is hard to see that the modular neural network "30%, 7.1%, and 5.9% of the weights of monolithic dense NNs for depths 1, 2, and 3" it would be nice to have a graph of the parameter counts, for each architecture.
Questions:

Line: 412 "Generalization on Seen Digit Combinations: Figure 7bb" the letter "b" is repeated. It should be Figure 7b.

**Questions:**

See weakness and in addition:

Could you explore more Hierarchical Tasks, it would be interesting to see how the networks perform on 4 and 5 depth tasks, and how this method scales?

Are you able to interpret what the reusable modules are doing and how this compares to the dense neural network?

---

> ### Author Response · Authors · 2024-11-23
>
> We sincerely thank the reviewer for their thoughtful comments and constructive feedback. Below, we address the weaknesses (numbered W1, W2, ..) and questions (numbered Q1, Q2) raised.
>
> **W1**: We will include a detailed walkthrough of the MNIST-based task in the appendix and revise the relevant sections in the main text to enhance clarity in the next version of the paper.
>
> **W2**: We appreciate the reviewer highlighting this point. In the next version, we will add a subsection to the Related Work section (see response to W1 for Reviewer jXZZ) that discusses how modularity improves interpretability, referencing prior works that have used modularity to reverse engineer and interpret trained networks.
>
> The interpretability of modular and modular-shared networks arises from their learned inter-module connectivity and functional specialization. Section 3.3 compares these learned features to the ground truth function graph. In cases where the ground truth is unknown, individual modules can be reverse-engineered and reassembled into a hierarchical structure to provide end-to-end interpretability. Since these modules are smaller and typically learn simpler sub-functions, the reverse engineering process is more straightforward.
>
> **MISC**: We will address the issues noted by the reviewer in the next version of the paper.
>
> **Q1**: We agree this is a valuable avenue for future exploration. However, due to limited time during the discussion, we prioritized addressing the additional experiments in the multi-task learning setting.
>
> **Q2**: While the question is not entirely clear, we have attempted to address it to the best of our understanding. Please see our response to W2 and Section 3.3, which detail the comparison between ground truth function graphs and the trained modular and modular-shared networks.
>
> The key difference between modular-shared and dense NNs lies in how modular-shared NNs align with the underlying task structure after training. Modular-shared NNs acquire a structure that mirrors the hierarchical composition of the task and learn sub-functions within their modules that correspond to the gates in the function graphs.
>
> **Note**: We urge the reviewer to also go through the newly added supplementary material pdf that presents experiments and results with multi-task learning.

---

> > ### Author Response · Authors · 2024-11-29
> >
> > Dear Reviewer,
> >
> > We hope this message finds you well. We wanted to kindly follow up to check if there are any additional questions or concerns regarding our previous response. If there are any points that require further clarification, we would be happy to address them.
> >
> > Your feedback is invaluable, and we appreciate your time and effort in reviewing our work.
> >
> > Thank you,

---

> > > ### Comment · Reviewer_RcFR · 2024-12-02
> > >
> > > Thank you for your responses; my understanding is that these will be addressed properly in the camera-ready version with acceptance due to spending more time addressing the other reviewers' concerns.
> > >
> > > I will maintain my score, as the work presented here is interesting, valuable and well-presented, albeit on simple problems.

---

### Official Review · Reviewer_AVSL · 2024-11-04

**Soundness:** 3
**Presentation:** 3
**Contribution:** 3
**Rating:** 3
**Confidence:** 4

**Summary:**

The paper explores how modular neural networks (NNs) can outperform traditional dense NNs, especially when tasks have an underlying hierarchical structure. The authors compare different types of NNs, from dense and sparse monolithic models to modular networks that incorporate various levels of task-specific structural information. They show that modular networks, especially those designed to reuse modules or with fixed connectivity, perform better in terms of learning efficiency and generalization, particularly when data is limited.

**Strengths:**

1.The paper provides a detailed comparison of several NN architectures and clearly shows when modular networks have an advantage.

2. The paper takes a solid scientific approach exhaustively covering the different ways in which hierarchical modularity can be achieved (random, learnable, fixed).

3.The paper is backed with controlled & thorough experiments on simple tasks such as Boolean functions and the MNIST dataset, where the underlying modularity is known.

**Weaknesses:**

A major limitation with prior work in hierarchical modular architectures is that such approaches haven't been shown to be effective on more complex tasks. In general, it seems that that, at scale, the inductive bias of hierarchical modularity may limit the models and choosing instead to use a less-constrained architecture, trained with more data, leads to better efficiency. This is the key message of the bitter lesson. While this systematic study is indeed very helpful in confirming when simple hierarchically modular inductive biases can enable more-efficient learning and / or better generalization, the challenge of understanding if this approach can scale still limits the practicality of this work.

**Questions:**

Covered in weaknesses.

---

> ### Author Response · Authors · 2024-11-23
>
> We acknowledge and share the reviewer’s concerns about the scalability of hierarchical modularity. As noted, “at scale, the inductive bias of hierarchical modularity may limit the models, and choosing a less constrained architecture trained with more data could lead to better efficiency.”
>
> The primary objective of this work, however, is to systematically explore how neural networks can benefit from the knowledge of specific properties of task structure. By employing architectures with clearly defined characteristics (e.g., modularity, module reuse), we isolate and assess the influence of these factors.
>
> While extending these findings to more complex tasks and larger models is an important direction, such an exploration would need to be tailored to the context of specific tasks and architectures. The central question remains: “What degree of knowledge about the specific large-scale task can be incorporated into the network architecture to improve generalization performance and training efficiency?”
>
> Prior work has employed general hierarchical and modular networks for various tasks without incorporating additional task-specific knowledge. However, as we demonstrate, modular networks alone do not necessarily yield better generalization or training efficiency—even for Boolean functions. Only with module reusability do we observe significant improvements. For example, Neural Module Networks [1] proposed a hierarchical and modular architecture with pre-defined task-specific modules, highlighting that incorporating domain-specific prior knowledge may be essential for tackling large-scale tasks effectively.
>
> **Note**: We urge the reviewer to also go through the newly added supplementary material pdf that presents experiments and results with multi-task learning.
>
> [1]: Andreas, Jacob, et al. "Neural module networks." Proceedings of the IEEE conference on computer vision and pattern recognition. 2016.

---

> > ### Comment · Reviewer_AVSL · 2024-11-24
> >
> > Thank you for your response. I have read the rebuttal and gone through the newly-added appendix.
> >
> > I am interested in the direction of modularity in neural networks and there has been prior work in this direction [1 and its follow up works]. However, while modularity in terms of composing networks trained for specific tasks has been effective [1], the modularity covered in this paper, i.e., where networks are trained end-to-end, despite being ideologically appealing, hasn't shown much practical benefit.
> >
> > I will maintain my score since I believe the analysis provided here is still useful to the community, but I would urge the authors to come up with more convincing arguments to motivate the modularity they study in more practical use cases.
> >
> > References:
> > [1] Neural Module Networks (https://openaccess.thecvf.com/content_cvpr_2016/papers/Andreas_Neural_Module_Networks_CVPR_2016_paper.pdf)

---

### Official Review · Reviewer_jXZZ · 2024-11-05

**Soundness:** 3
**Presentation:** 2
**Contribution:** 2
**Rating:** 3
**Confidence:** 3

**Summary:**

This paper investigates how modular neural networks can outperform traditional dense networks by varying the degrees of structural knowledge used. They compare monolithic and hierarchically modular neural networks on boolean functions and MNIST-based modular tasks. They analyze generalization performance, training efficiency (FLOPs required for training), and factors that influence generalization in modular networks. The experiment results show that incorporating structural knowledge, especially via module reuse and fixed connectivity, helps improve learning efficiency and generalization.

**Strengths:**

* The experiment results are comprehensive across various architectural designs with varying levels of structure knowledge, comparing their generalization performance and training efficiency.
* They discover specific design choices for modular networks that help generalization for boolean functions and MNIST-based modular tasks.

**Weaknesses:**

* The paper needs much better contextualization relative to prior works. Each architectural choice—such as learning inter-module connectivity and modular sharing—should be supported with citations of relevant papers. Are these established approaches for designing modular networks, or are they introduced by the authors? Furthermore, are Boolean function and MNIST-based tasks standard evaluation protocols for modular networks?
* The paper lacks proper citations in several sections. The design choice for the Boolean function and MNIST-based task follows the approach in [1] and should be cited accordingly.
* Their experiments are limited to MLPs on boolean functions and MNIST, which are relatively simple architectures and tasks. It is unclear how the findings of this paper would generalize to more complex architectures and problem settings.

[1] Neural Sculpting: Uncovering hierarchically modular task structure in neural networks through pruning and network analysis, NeurIPS 2023.

____
Post rebuttal comments:

Thank you for your detailed response and the newly added supplementary material.

I believe the authors investigate an important question. However, the authors mainly investigate architectural choices specifically tailored to simple tasks, essentially toy setups, which limits the practical relevance of the conclusions for the broader community. While the authors provide the multi-task experiment in the rebuttal, it remains in the domain of Boolean functions.

To reduce variability in the review scores, I lower my score to 3: reject.

**Questions:**

It is mentioned in the weakness section.

---

> ### Author Response · Authors · 2024-11-23
>
> We sincerely thank the reviewer for their constructive comments. Below, we address each identified weakness (numbered W1, W2, ..)
>
> **W1**: We recognize the need for better contextualization relative to prior work and will include a dedicated "Related Works" section in the next version of the paper, along with revisions to the introduction.
>
> The high-level architectural choices in our work are not directly tied to specific prior architectures but are designed as a systematic framework for incorporating varying levels of structural information about the underlying task. However, certain implementation details, such as module input selection and module sharing, are adaptations of established mechanisms from prior works [1,2], which we have simplified or tuned for the tasks considered.
>
> Our focus on modular and hierarchical tasks aligns with the definitions in [3]. Similar to [3], we assume the structure of a single task to be static and fixed, and we use comparable tasks. In the multi-task setting, these structures adapt based on the input task identifier, as detailed in the newly added supplementary material pdf.
>
> **W2**: We appreciate the reviewer highlighting the lack of citations in several sections. We will address this by adding the relevant references in the next version of the paper.
>
> **W3**: We acknowledge that the experiments are limited to MLPs on Boolean functions and MNIST-based tasks, which are relatively simple. However, the primary goal of this work is to systematically examine various aspects of task structure and identify how neural networks can benefit from these properties. By using architectures with well-defined characteristics (e.g., modularity, module reuse), we isolate and evaluate the influence of specific factors.
>
> While extending these findings to more complex tasks and larger models is an important future direction, such exploration would need to be conducted in the context of specific tasks and architectures. A general study addressing this broader scope would fall outside the focus of this paper.
>
> **Note**: We urge the reviewer to also go through the newly added supplementary material pdf that presents experiments and results with multi-task learning.
>
> [1] - Goyal, Anirudh, et al. “Neural Production Systems: Learning Rule-Governed Visual Dynamics.” CoRR, abs/2103.01937 (2022)
>
> [2] - Ostapenko, Oleksiy, et al. “Attention for compositional modularity.” NeurIPS'22 Workshop on All Things Attention: Bridging Different Perspectives on Attention. 2022
>
> [3] - Malakarjun Patil, Shreyas, et al. "Neural Sculpting: Uncovering hierarchically modular task structure in neural networks through pruning and network analysis." NeurIPS 2023.

---

> > ### Author Response · Authors · 2024-11-29
> >
> > Dear Reviewer,
> >
> > We hope this message finds you well. We wanted to kindly follow up to check if there are any additional questions or concerns regarding our previous response. If there are any points that require further clarification, we would be happy to address them.
> >
> > Your feedback is invaluable, and we appreciate your time and effort in reviewing our work.
> >
> > Thank you,

---

### Comment · Area_Chair_XRnk · 2024-11-24

Dear Reviewers,

This is a gentle reminder that the authors have submitted their rebuttal, and the discussion period will conclude on November 26th AoE. To ensure a constructive and meaningful discussion, we kindly ask that you review the rebuttal as soon as possible and verify if your questions and comments have been adequately addressed.

We greatly appreciate your time, effort, and thoughtful contributions to this process.

Best regards,
AC

---

### Meta-Review · Area_Chair_XRnk · 2024-12-16

**Metareview:**

This work shows that modular networks can be more efficient learners when their connectivity pattern matches that of the task to be solved. They focus on Boolean operations on the MNIST dataset. During the rebuttal they also include a multi-task setting.

As perceived by reviewers:

* The main strength is that this work achieves solid results showing that matching the structure of the data makes learning more efficient.
* The main weakness is that it does so on a very limited and constrained setup that might not scale to more complex situations.

Following reviewers’ guidance, I recommend to reject this work from ICLR 2025.

**Additional Comments On Reviewer Discussion:**

* Reviewers jXZZ, AVSL, TwAX, recommended a score of 3: reject since they believe the proposed setup would not transfer to more complex tasks.
* Reviewer RcFR finds value in this work but during the AC/reviewer discussion they tended towards agreement with jXZZ, AVSL, TwAX and thus would not fight for acceptance.

Following reviewers’ guidance, I recommend to reject this work from ICLR 2025.

---

### Decision · Program_Chairs · 2025-01-22

Reject